# Hot luminescence from single-molecule chromophores electrically and mechanically self-decoupled by tripodal scaffolds

Vibhuti Rai [1,7], Nico Balzer[2,7], Gabriel Derenbach[1], Christof Holzer [3], Marcel Mayor [1,2,4,5] ✉, Wulf Wulfhekel [1,6], Lukas Gerhard[1] ✉ & Michal Valášek [2] ✉

Control over the electrical contact to an individual molecule is one of the biggest challenges in molecular optoelectronics. The mounting of individual chromophores on extended tripodal scaffolds enables both efficient electrical and mechanical decoupling of individual chromophores from metallic leads. Core-substituted naphthalene diimides fixed perpendicular to a gold substrate by a covalently attached extended tripod display high stability with well-defined and efficient electroluminescence down to the single-molecule level. The molecularly controlled spatial arrangement balances the electric conduction for electroluminescence and the insulation to avoid non-radiative carrier recombination, enabling the spectrally and spatially resolved electroluminescence of individual self-decoupled chromophores in a scanning tunneling microscope. Hot luminescence bands are even visible in single self-decoupled chromophores, documenting the mechanical decoupling between the vibrons of the chromophore and the substrate.

Control of the molecular arrangement and spatial orientation of light-emitting molecules on metallic surfaces is of crucial importance for the construction of artificial photonic devices such as organic light-emitting diodes or single photon sources for quantum optical technologies[1,2]. When single molecules are to be used as electrically powered photon sources, their coupling to the electrodes has to be optimized: Light emission from the chromophore is hampered when their orbitals are too strongly hybridized with the metallic leads due to quenching of the chromophore's excited state by the metal's free electrons, but it is not functional either when electrically isolated so that no current passes through. From an experimental view-point, the vacuum gap of a scanning tunneling microscope (STM) provides a tuneable electric contact to single molecules[3–6]. However, the coupling to the second lead, provided by the metal substrate, has to be reduced, and this is typically achieved by a thin insulating layer with a large band gap between the metal surface and the chromophores[7–16]. A purely molecular approach of integrating the insulating spacer between chromophore and substrate by molecular design is less explored in spite of its conceptual attractiveness and application potential. Particularly appealing are multipodal platforms mounting the optically active molecular subunit not only at a well-defined distance above the metal surface, but also controlling its spatial arrangement[17,18]. A variety of molecular architectures combining footing platforms with exposed chromophores have been developed and integrated in STM-based electroluminescence experiments[19–24]. The diversity of observed emission features in the individual experiments and/or the limited stability of the emitting junctions hint at the challenges that remain in competition with experimental set-ups that rely on insulating layers.

A key to a spatially well-defined arrangement of the mounted chromophore is the molecular design of the platform scaffold.

[1]Institute for Quantum Materials and Technologies, Karlsruhe Institute of Technology (KIT), Karlsruhe, Germany. [2]Institute of Nanotechnology, Karlsruhe Institute of Technology (KIT), Karlsruhe, Germany. [3]Institute of Theoretical Solid State Physics, Karlsruhe Institute of Technology (KIT), Karlsruhe, Germany. [4]Department of Chemistry, University of Basel, Basel, Switzerland. [5]Lehn Institute of Functional Materials, Sun Yat-Sen University (SYSU), Guangzhou, China. [6]Physikalisches Institut, Karlsruhe Institute of Technology (KIT), Karlsruhe, Germany. [7]These authors contributed equally: Vibhuti Rai, Nico Balzer. ✉e-mail: marcel.mayor@unibas.ch; lukas.gerhard@kit.edu; michal.valasek@kit.edu

Numerous parameters have to be optimized like (i) the strength of the interaction between platform and substrate, (ii) the lateral dimension of the platform and (iii) its structural rigidity and integrity[17,18]. The strength of the platform immobilization on the substrate depends on the nature of the interaction between them. Flat delocalized molecular subunits tend to interact with the planar substrate by physisorption with a moderate binding energy enabling lateral self-assembly to periodically arrange molecules in monolayers, but reducing the probability of separated individual molecules required for single molecule studies[19]. An appealing alternative is the decoration of the platform with thiol anchor groups forming covalent sulfur-gold bonds with the substrate[25]. The thereby chemisorbed system has a substantially increased binding energy enlarging the probability for separated single molecules. The lateral dimension of the platform turns out to be of crucial importance. In a first attempt with a tripodal platform of smaller dimension, the physisorption of the mounted chromophore was favored over the chemisorption of the foot structure, resulting in a flat arrangement of the chromophore on the surface, additionally stabilized by two of the three anchor groups forming covalent bonds[23,24]. An effective strategy to increase the interaction of the foot structure with the substrate is to enlarge its size[26], with the additional advantage that the distance between perpendicularly mounted chromophore is increased, avoiding quenching by interchromophore stacking. Finally, structural rigidity of the platform is required to obtain control over the spatial arrangement of the emitting chromophore. With a central $sp^3$-hybridized carbon atom as a mounting point, enlarging the π-systems of the three footing substituents increases their chemisorption and thereby planarizes the entire foot structure and forces the fourth substituent into a perpendicular arrangement. The crucial parameter of the electroluminescence experiment, the extent of coupling between the chromophore mounted as the fourth substituent and the substrate, becomes adjustable by its linking structure. The interaction between the mounted chromophore and the substrate depends on the

chemical nature of the linking structure, the distance and orientation, as the transition dipole moment of the chromophore should have substantial contributions perpendicular to the surface to favor strong plasmonic enhancement[13,27–29].

In this paper, we report single-molecule STM electroluminescence experiments with self-decoupled chromophores. Two model compounds (Fig. 1: **Tol-Tpd-sNDI**-Ac and **Tol-Tpd-nNDI**-Ac, with s/n indicating the heteroatom of the core substituent) were developed and synthesized for the experiment combining a 2,6-disubstituted naphthalene-1,4,5,8-tetracarboxdiimide (NDI) chromophore with a tripodal foot structure to control both their immobilization on and their interaction with the Au (111) surface. The tripodal platform **Tol-Tpd** with its thiol decorated tolane π-systems fixes the model compound firmly on the metal substrate and ensures an upright arrangement of the mounted chromophore. The phenylethynyl linker between the foot structure and the chromophore decouples the latter from the substrate. The model compounds were introduced into the ultrahigh vacuum (UHV) STM experiment with an in-house developed spray technique, which resulted in separated individual molecules, **Tol-Tpd-sNDI** or **Tol-Tpd-nNDI**, standing upright on their foot structures. Here, the low temperature STM based electroluminescence investigations show that the optical properties of the NDI chromophore at the single molecule level are preserved, including not only the tuneability by core substitution, but also resolved vibrational features and relatively high luminescence quantum yields. The integrity and stability of the immobilized molecule enables systematic STM-induced luminescence (STML) experiments including photon maps, which so far were only possible for molecules decoupled via insulating layers. In addition to the electrical decoupling, the tripodal scaffold also provides an impressive degree of mechanical decoupling of the chromophore, resulting in hot luminescence due to the increased lifetimes of the chromophores' vibrational modes. This chemical approach to self-decoupling is not limited to the presently reported NDI dyes, its modular assembly allows to alter the exposed chromophore to further

**Fig. 1 | Synthesis of the target tripodal molecules and their integration into the UHV STM experiment.** Reaction conditions: **a** CuI, Pd(PPh₃)₄, Et₃N, THF, 74% (**Tol-Tpd-sNDI**-TMSE) and 71% (**Tol-Tpd-nNDI**-TMSE); **b** AcCl, AgBF₄, CH₂Cl₂, 62% (**Tol-Tpd-sNDI**-Ac), 76% (**Tol-Tpd-nNDI**-Ac); **c** Tol-Tpd-sNDI-Ac or **Tol-Tpd-nNDI**-Ac as $10^{-4}$ mol L⁻¹ solution in CH₂Cl₂ sprayed through a pulse-valve (10 ms) onto the Au(111) substrate in UHV, followed by annealing to ≈373 K in UHV for about 1 h.

optimize light emitting properties of the molecular junction. Furthermore, the structural integrity and rigidity of the immobilized platform on the surface provides control over the orientation of the chromophore and thus also of its transition dipole moment.

## Results

### Molecular design and synthetic strategy

Naphthalene diimides are compact, planar, and stable chromophores with electronic and optical properties that can be adjusted by the core substituents[23,30], thus finding promising applications in optoelectronic devices[31]. The intention of the molecular design is to fix the chromophore upright on the substrate, minimizing its interaction with the surface. The NDI dyes were identified as the ideal optically active subunit due to their robustness, optical tunability, structural integrity, and synthetic accessibility. In the first series of NDIs mounted on a smaller tripodal platform based on tris(*meta*-sulfanylphenyl) methane[23], the physisorption of the chromophore efficiently competed with the intended fixation by the three foot subunits of the tripod, pointing at the subtle balance between the surface interactions involved[23]. As the size of the chromophore in the presently reported second series remains the same, the foot structure was enlarged by three tolane subunits, increasing its lateral dimension and thus also its interaction with the surface. The strategy to increase the likelihood of an upright orientation by enlarging the footprint of the tripod was already reported by Yao and Tour[26]. The combination of the three physisorbed tolane $\pi$-systems with the chemisorbed thiol groups should firmly fix the foot structure to the surface and thereby ensure an upright orientation of the fourth substituent. The NDI chromophores are mounted through the rigid ethynylphenyl spacer, lifting the chromophore above the surface and decoupling it electronically, which also leads to a pronounced mechanical decoupling, at least in the low temperature STM experiment. After these considerations, the proposed tripodal chromophores are based on an extended tripodal architecture with three acetyl-protected thiol anchors in the *meta* positions of the three tolane arms and NDI chromophores with 2,4,6-trimethylphenylsulfanyl (**sNDI**) and pyrrolidine (**nNDI**) core-substitution. The final steps of the assembly of the target molecules **Tol-Tpd-sNDI**-Ac and **Tol-Tpd-nNDI**-Ac are outlined in Fig. 1. After multistep synthesis of tolane-based tripod **Tol-Tpd**-H and both core-substituted NDI chromophores (see Methods and Supplementary Methods), Sonogashira reaction was used to couple **sNDI**-I and **nNDI**-I molecules with the modular platform **Tol-Tpd**-H to afford precursors **Tol-Tpd-sNDI**-TMSE and **Tol-Tpd-nNDI**-TMSE. Final transprotection of the thiols was successfully performed using AgBF$_4$ and acetyl chloride, yielding the desired acetyl masked target structures **Tol-Tpd-sNDI**-Ac and **Tol-Tpd-nNDI**-Ac. Acetyl protecting groups, which prevent the accidental formation of oligo- and/or polymeric species by the formation of disulfides, e.g. by molecular oxygen, turned out to be particularly useful and, furthermore, can be easily deprotected either in situ upon binding to the gold surface or chemically by cleaving agents.

### STM-induced luminescence of molecular chromophores

For our low-temperature STM studies, the NDI model compounds **Tol-Tpd-sNDI**-Ac and **Tol-Tpd-nNDI**-Ac were deposited as dichloromethane solution onto an atomically clean Au(111) surface by an in-house developed spray technique, which was optimized and successfully applied for tripod model compounds which cannot be sublimed, since their degradation temperatures lie below their sublimation temperatures[23,24,32,33]. Subsequent thermal annealing before cooling down to measurement temperatures promotes deprotection of the thiol anchors and the intended upright configuration with the chromophore pointing away from the substrate (details of the sample preparation can be found in Methods). Figure 2a, b shows the schematic of the desired upright configuration of the **Tol-Tpd-sNDI** and

**Tol-Tpd-nNDI** junctions with the tripodal scaffold anchored to the Au(111) substrate, keeping the chromophores (**nNDI** and **sNDI**) away from the metal surface. Typical STM images of such samples (see Fig. 2c, d) show individual high objects with apparent height/width of about 850 pm/2 nm (depending on the scanning parameters), and are compatible with the desired upright configuration.

To measure the STML signal from **Tol-Tpd-nNDI** junctions, the tip was placed on top of an immobilized molecule (see the inset in the top panel of Fig. 2e) in order to inject electrons into the molecule while the STML spectrum was recorded[34]. The spectrum shows a sharp high-intensity peak $Q_n$ at ≈1.96 eV (see highlighted blue shaded region in Fig. 2e) in combination with several low-intensity peaks at lower energies, characteristic for molecular luminescence[7,35]. A similar situation is found for **Tol-Tpd-sNDI** (Fig. 2e, bottom) with the main emission line at 2.25 eV demonstrating the chemical tuneability of the emission line by core substitution. The fine structure of the spectrum will be discussed later in this report. The principal peaks agree perfectly with the ensemble broadened photoluminescence spectra in solution (blue and red dotted lines in Fig. 2e). They also agree with the accuracy of time-dependent density functional theory (TD-DFT) calculations performed using the TMHF local hybrid functional (TD-DFT locates the lowest adiabatic excitation energies at 2.21 eV for **Tol-Tpd-nNDI** and 2.47 eV for **Tol-Tpd-sNDI** deviating only by a slight but constant blue shift of 0.2 eV). The calculated absorption spectra, corrected for the adiabatic energy shift, are shown in Suppl. Fig. 6 together with the natural transition orbitals (NTOs) for both molecules which are mainly located at the central NDI moiety (for the optimized structures of the target molecules see Supplementary Data 1, 2). Note that with the design of the molecules, also the orientation of the transition dipole moment with respect to the *z*-axis (normal to the surface) is fixed (see Suppl. Fig. 7 and Suppl. Table 3). Especially when oriented perpendicular to the surface, the quantum efficiency can be further boosted[13,27–29].

For **Tol-Tpd-nNDI**, we observe high quantum yields of up to ≈2.7 × 10$^{-3}$ photons per electron (corrected for detector efficiency, see Suppl. Fig. 2), which indicates that the tripodal scaffold provides sufficient electronic decoupling of the chromophore so that it escapes the fluorescence quenching caused by the metal surface underneath. The fact that the photoluminescence (PL) spectrum in solution (see Suppl. Fig. 9) matches very well the STML spectrum indicates that the molecule in the **Tol-Tpd-nNDI** junction is in its neutral state. Note that the STML efficiency of **Tol-Tpd-sNDI** junctions (≈1 × 10$^{-6}$ photons per electron) is three orders of magnitude lower than those of **Tol-Tpd-nNDI** junctions, which is consistent with the relative fluorescence quantum yields observed for the corresponding chromophores in solution[23] and our gas phase calculations (see Suppl. Figs. 7, 8, and Suppl. Table 3), further demonstrating the balanced decoupling of the chromophore in the experiment, as the limits of its optical performance are dominated by its chemical structure.

Also, in terms of reproducibility the presently reported approach with tailor-made model compounds goes beyond previous attempts to self-decouple chromophores[20,21,23,24], as very similar STML spectra were recorded with numerous molecular junctions (see Suppl. Fig. 3) without the need of any tip-induced manipulation of the molecule[24,36,37].

To understand the origin of the emission lines of **Tol-Tpd-nNDI** and **Tol-Tpd-sNDI**, we measured the integrated photon counts of the main lines $Q_n$ and $Q_s$ as a function of the applied sample bias $U$ (see Fig. 3a). $Q_n$ and $Q_s$ are absent for negative $U$ (the small photon counts in the negative bias are of plasmonic origin). For both **Tol-Tpd-nNDI** and **Tol-Tpd-sNDI** emission of $Q_n$ and $Q_s$ is observed as soon as the energy of the tunneling electrons equals the energy of the emission lines $Q_n$ and $Q_s$. Based on previous studies, this onset of STML, once the corresponding excitation energy is provided by the tunneling electrons, is an indication that the chromophore is excited by inelastic energy transfer in the plasmonic junction[37–39]. This is in full agreement with STML experiments at different tunneling currents (see Suppl. Fig. 5).

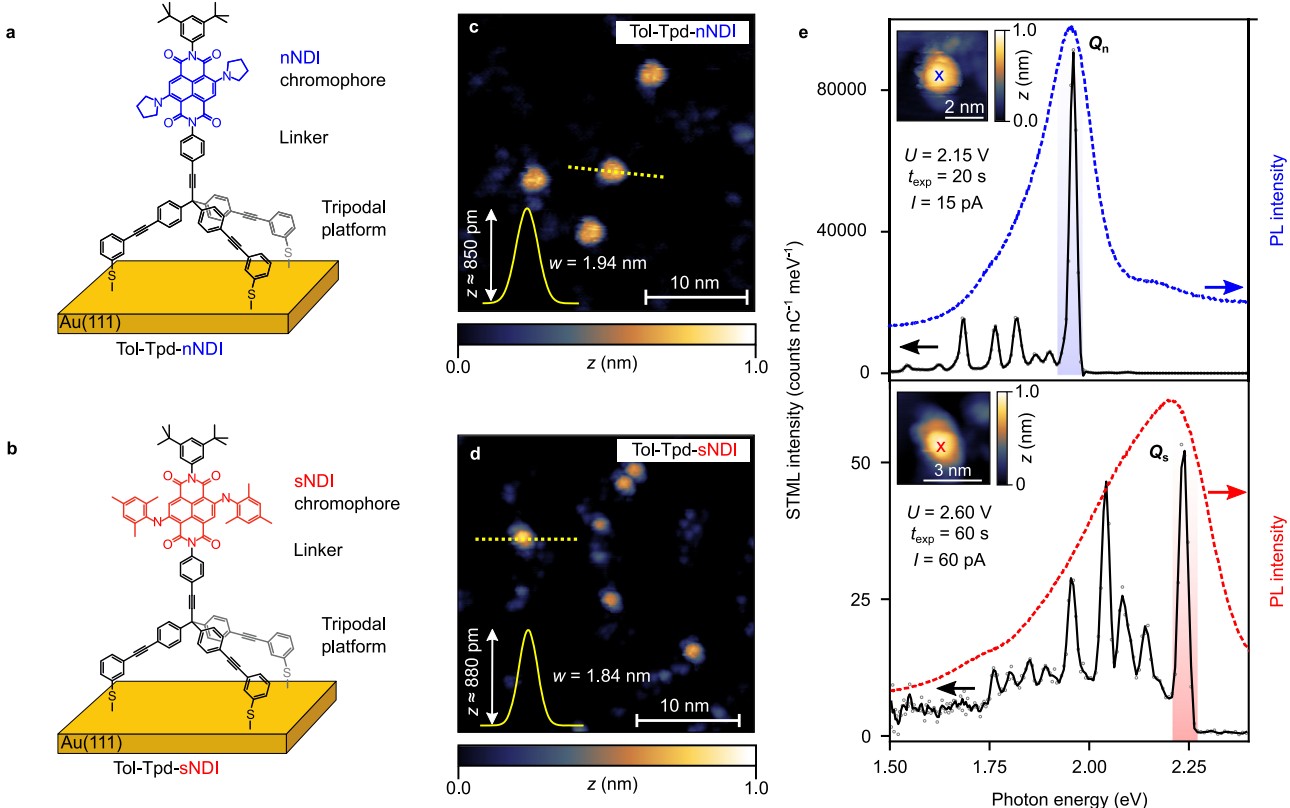

**Fig. 2 | Sketch of the immobilized chromophores (a-b), their STM topographies (c, d), and their STML spectra (e). a, b** Sketch of the immobilized tripodal structure exposing the NDI chromophore in the upright configuration. **c, d** STM topographies of **Tol-Tpd-nNDI** ($U = 1.5$ V, $I = 2.0$ pA) and **Tol-Tpd-sNDI** ($U = 2.0$ V, $I = 2.0$ pA). The yellow solid lines depict a Gaussian peak fitted to the profile along the dashed yellow lines with the peak height ($z$) and full width at half maximum ($w$) indicated. Apparent height ($z$) in topographies from 0 to 1.0 nm (black to white) in **c**–**e**. **e** STML spectrum (top panel, in black, $U = 2.4$ V, $I = 15$ pA, $t_{exp} = 20$ s, main line $Q_n$ marked by blue shade) recorded by placing the STM tip on top of an individual

**Tol-Tpd-nNDI** (position is marked with a blue cross in the top panel inset) with the photoluminescence spectrum plotted in blue dotted line. STML spectrum (bottom panel, in black $U = 2.6$ V, $I = 60$ pA, $t_{exp} = 60$ s, main line $Q_s$ marked by red shade) recorded on top of an individual **Tol-Tpd-sNDI** (position is marked with a red cross in the bottom panel inset) with PL spectrum plotted in red dotted line. PL spectra for both the molecules were recorded in dichloromethane with a concentration of 25 μM at ambient temperature[23]. STML spectra are corrected for detector efficiency (see Suppl. Fig. 2).

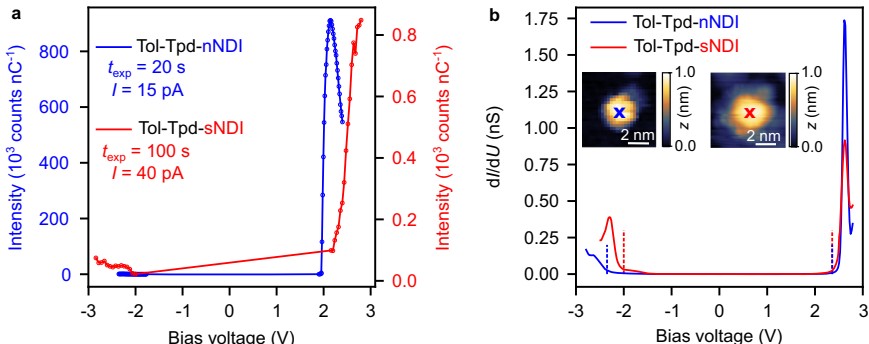

**Fig. 3 | Energetics of the $Q_n$ and $Q_s$ emission lines and STS. a** Intensity of $Q_n$ (blue line and scale) and $Q_s$ (red line and scale) measured on **Tol-Tpd-nNDI** and **Tol-Tpd-sNDI** molecules as function of the applied bias voltage. **b** Differential conductance d$I$/d$U$ of **Tol-Tpd-nNDI** (blue) and **Tol-Tpd-sNDI** (red) on the center of the

molecules as indicated by blue and red crosses, respectively. Dashed lines indicated the onset of the resonances as discussed in the text. Apparent height ($z$) in topographies from 0 to 1.0 nm (black to white).

To further confirm this, we also measured the differential conductance spectra (d$I$/d$U$) on both molecules (see Fig. 3b). For both molecules, sharp negative ion resonances (NIRs) are observed with an onset of $U = 2.36$ V, while in the negative $U$, positive ion resonances (PIRs) are observed at −2 V and −2.4 V for **Tol-Tpd-sNDI** and **Tol-Tpd-nNDI**, respectively (see red and blue dotted lines in Fig. 3b).

For **Tol-Tpd-nNDI**, the NIR with an onset of $U = 2.36$ V appears clearly above the threshold voltage of 1.95 V for the onset of the main emission line $Q_n$ (with an energy of 1.95 eV), excluding excitation via sequential charging and discharging of the molecule at voltages up to 2.36 V. The onset of an additional excitation mechanism via sequential charging and discharging of the molecule would result in a step-wise

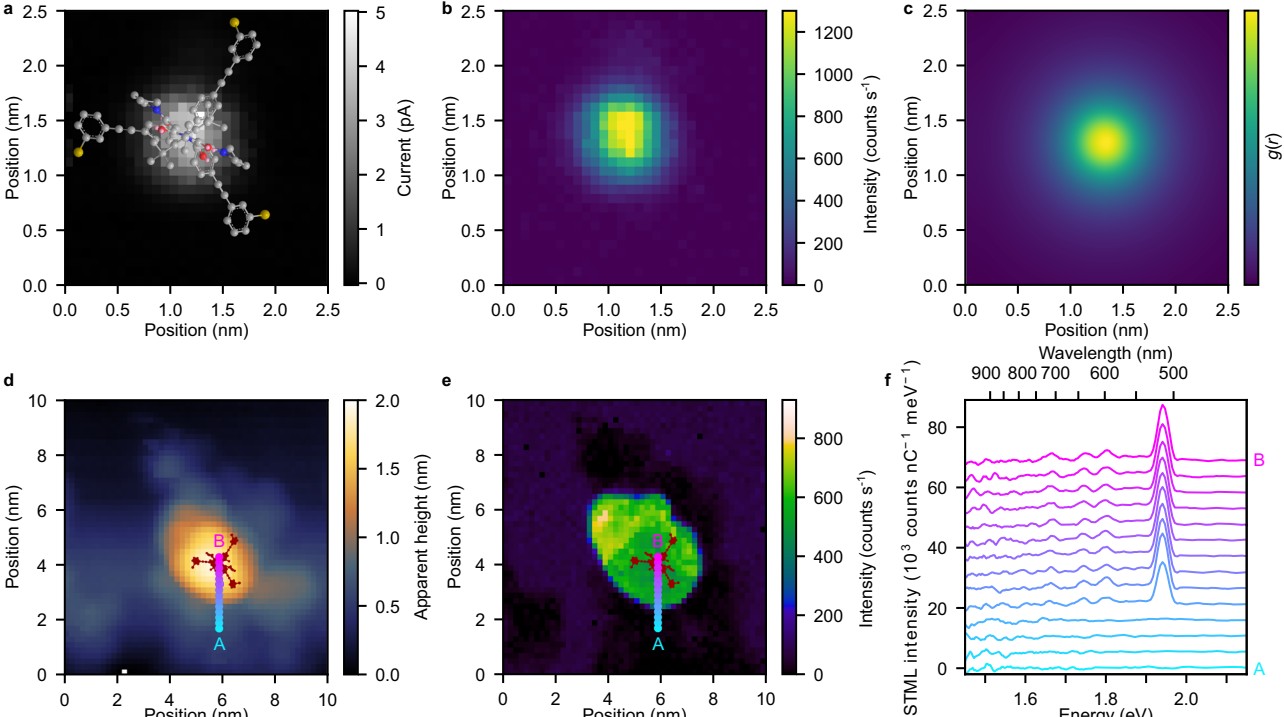

**Fig. 4 | Spatial distribution of light emission. a** Current map of a single **Tol·Tpd-nNDI** junction at a constant height mode ($U = 2.4$ V) representing the outermost orbitals. The top view of the standing upright molecular structure is superimposed to scale. C/N/O and S atoms are marked in gray/dark blue/red and yellow, respectively. **b** Simultaneously recorded photon map showing the spatial distribution of the integrated intensity of the $Q_n$ emission line ($1.907 - 1.983$ eV). Tunneling parameters for recording the photon map are $U = 2.4$ V, $t_{exp} = 2$ s. **c** Simulated photon map (for details see Suppl. Fig. 6 and corresponding text). **d** Topography of a single **Tol·Tpd-nNDI** in constant current mode ($U = 2.4$ V, $t_{exp} = 6$ s, $I = 1$ pA, 50×50 pixels). The outline of the molecular structure is shown in black in **d** and **e**. **e** Simultaneously recorded photon map showing the intensity of the $Q_n$ line. **f** Individual spectra extracted from the photon map along the line from A to B as indicated with the cyan-magenta dots in **d** and **e**.

increase of the photon emission rate[12], which we do not observe for voltages up to 3 V (for **Tol·Tpd-nNDI**, see Suppl. Fig. 4). For **Tol·Tpd-sNDI**, the onset of NIR is very close to the energy of $Q_s$ line and it is nontrivial to make a decisive statement about the excitation mechanism. The sharpness of the NIR peaks and the clear energy gap between the PIR and the NIR peaks suggest that the molecules are electronically well decoupled.

In addition, the mechanical stability of the **Tol·Tpd** platform allows us to perform spatially resolved STML measurements, which has not been reported in previous work on such self-decoupled molecules and clearly shows the high stability of the molecular junction. Figure 4a shows a current map of a **Tol·Tpd-nNDI** junction as a function of $x$ and $y$ coordinates over a grid of 30×30 pixels at a fixed $z$ position of the tip (constant height mode). Simultaneously, STML spectra are recorded at each pixel and the integrated photon count of the $Q_n$ line is plotted (Fig. 4b). The almost circular symmetric distribution of the light emission is expected for an upright molecule and is in full agreement with a simulated photon map (see Fig. 4c). In order to measure light emission spectra over a larger area of the molecule and on the substrate close to the molecule, we recorded a photon map at constant current which requires adjustment of the tip-sample distance. Thus, as expected for an upright molecule, both the topography (Fig. 4d) and the area of light emission (Fig. 4e) are heavily influenced by the shape of the apex of the tip and do not show any sub-molecular features. These photon maps show a smooth transition in the intensity as the tip moves closer and farther away from the immobilized molecule, proving the stability of the molecular junction. In contrast to flat molecules, for the upright molecule discussed here, the current is always injected into the apex of the molecule and there is a sharp transition from molecular luminescence

to the weak plasmonic light emission when moving the tip away from the molecule (Fig. 4f).

In contrast to experiments relying on insulating layers, electronic decoupling of the chromophore via a tripodal scaffold also implies a certain degree of mechanical decoupling as the chromophore part is not in direct contact with the substrate. Therefore, the side bands in the STML spectra of a **Tol·Tpd-nNDI** junction were investigated with higher spectral resolution using a narrower entrance slit of the spectrometer (see Suppl. Fig. 3) at increasing bias voltages.

The STML spectrum recorded at $U = 1.98$ V, i.e. at a bias voltage just above the photon energy of the main line $Q_n$, shows a fully developed spectrum together with the lower energy peaks (LE-band) (see blue spectrum in Fig. 5a). We do not observe any peak at higher energy than the $Q_n$ line. As the bias voltage is increased further (black spectrum in Fig. 5a), several peaks at higher energies (HE-band) occur. These HE-band peaks are nearly mirror symmetric to the LE-band peaks with respect to the $Q_n$ line (see the dark blue, green, red and yellow shaded regions), which suggests that the same vibrational modes are involved. In full agreement with the literature[7,35,38,40–42], the above observations can be explained in the following way: The main line results from transitions from the first electronically excited state ($S_1$) to the electronic ground state ($S_0$) without a change in the vibrational state ($\nu$): $S_1 \nu'_n$ - $S_0 \nu_m$, where $n = m$ are positive integers indicating the vibrational quantum number. Electronic transitions that go along with an increase of the vibrational quantum number lead to the LE-band ($S_1 \nu'_n$ - $S_0 \nu_m$, where $n < m$, see the left inset in Fig. 5a). Transitions from a higher vibrational level to a lower vibrational level lead to the HE-band ($S_1 \nu'_n$ - $S_0 \nu_m$, where $n > m$, see the right inset in Fig. 5a). Note that, since the HE-band peaks are not excited together with the main $Q_n$ line, but only when the tunneling electrons have the energy of the

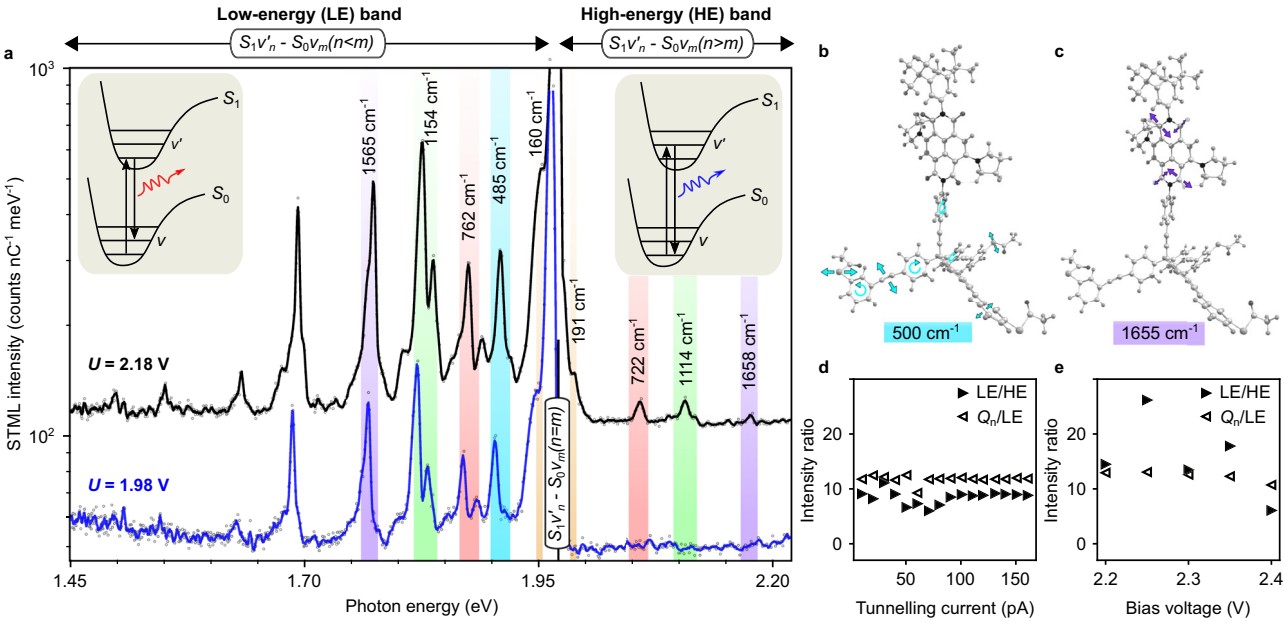

**Fig. 5 | Vibronic spectroscopy of Tol-Tpd-nNDI. a** Emission spectra recorded on a single **Tol-Tpd-nNDI** junction at two different bias voltages are shown in blue ($U = 1.98$ V, $I = 25$ pA, $t_{exp} = 500$ s) and in black ($U = 2.18$ V, $I = 25$ pA, $t_{exp} = 500$ s). Nearly mirror-symmetric peaks in LE-band and HE-band are marked with blue, green, red, and yellow shaded regions, and their energy shift from the $Q_n$ line are given in cm⁻¹. Energies of vibrational satellites are obtained from a Lorentzian fit. The cyan-shaded line of the LE band has no counterpart in the HE band. The left inset depicts transitions responsible for LE-band peaks where the excited system relaxes to vibrational excited states of $S_0$. The right inset depicts transitions leading to the HE-band peaks where the system de-excites directly from the excited vibrational states of $S_1$. **b, c** Illustration of a typical skeletal vibration at 500 cm⁻¹ **b**, and a typical imide mode at 1655 cm⁻¹ **c** calculated for the molecule in the gas phase. **d** Ratio of the intensities of the LE and HE ($Q_n$ and LE) lines of the imide mode at around 1600 cm⁻¹ as a function of the tunneling current **d** and the applied bias voltage **e**.

corresponding transition (see Suppl. Fig. 4), the excitation process is not involving annihilation of vibrons (which cause anti-Stokes lines). Such transitions involving non-thermally excited vibrations are known as hot luminescence (HL)[38,40,43] and present a violation of Kasha's rule[44]. HL has been observed in previous STML experiments by Dong et al.[38] in molecular multilayers and by Chong et al.[40] in molecular wires suspended between a metal surface and the STM tip, but has not been observed in the typical STML experiments with single molecules adsorbed on thin insulating layers[8,9,11,20], except in the case of perylene-3,4,9,10-tetracarboxylic dianhydride (PTCDA) molecules on NaCl[45,46]. HL occurs when the time of the photonic de-excitation is similar or shorter compared to vibrational de-excitation, which is typically not the case (Kasha's rule). There are two possible scenarios for this to happen: (1) The photonic de-excitation is drastically accelerated by coupling to gap plasmon-polaritons[47]. (2) The vibrational de-excitation is suppressed due to mechanical decoupling. Scenario 1, which has been used to rationalize previous observations of HL[38,40,47], is an effect of low energy selectivity, as is reflected in the broad gap-plasmon spectrum (see Suppl. Fig. 2b). It thus acts on all transitions in that energy range. As can be seen in Fig. 5a, four vibrational peaks on the LE side have a counterpart in the HL, but the LE-peak at 485 cm⁻¹ clearly does not, in spite of a good match to the plasmonic spectrum. This excludes the possibility of a cavity enhanced photon emission rate as the only explanation for the HL peaks. To test hypothesis 2 of mechanical decoupling, we calculated the vibrational spectrum of **Tol-Tpd-nNDI** to identify the nature of the vibrational modes that contribute to the different lines in the experimental spectrum (see Suppl. Tables 1, 2). In particular, we found that the line at 485 cm⁻¹ is composed of de-localized skeletal vibrations (Gerüstschwingungen) with a significant contribution of the foot structure (see Fig. 5b), so it can couple to the substrate and the vibrationally excited state is de-excited before the photon is emitted (following Kasha's rule). The other colored lines that appear in the HL have at least significant contributions from decoupled modes and in particular, we can clearly identify

the characteristic C=O (in imide) modes as the origin of the line in the range of 1600–1660 cm⁻¹ (see Fig. 5c; for details, see Suppl. Tables 1, 2, and Supplementary Movies 1-5 showing animations of the vibrational modes at the five energies indicated in Fig. 5a). In order to quantify this selective behavior, we estimated the experimental Franck-Condon factors (FCF) from the intensities of the $Q_n$ line and its vibrational satellites. The main line is due to a transition without change of the vibrational quantum number in two steps from the electronic ground state to the electronically excited state and back (i.e., $n = 0$ to $m = 0$ and $m = 0$ to $n = 0$ vibrational modes), involving the corresponding $FCF_{00}{}^2$. The lines on the LE side also involve excitation without change of the vibrational quantum number ($FCF_{00}$), but are followed by an electronic de-excitation to a vibrationally excited state ($FCF_{01}$). Thus, the ratio of the intensities of the two lines $Q_n$/LE is given by $(FCF_{00})^2/(FCF_{00} \times FCF_{01}) = FCF_{00}/FCF_{01}$. For the lines on the HL side, both the electronic excitation and de-excitation involve a change in the vibrational quantum number, i.e., $FCF_{01}$ and $FCF_{10}$, which can be approximated to be identical. With this, also the intensity ratio of LE/HE = $(FCF_{00} \times FCF_{01})/(FCF_{10} \times FCF_{01}) \approx FCF_{00}/FCF_{01}$. We tested this identity for the C=O line and show both ratios as a function of voltage and current in Fig. 5d, e. Indeed, the two ratios show very similar values, further supporting our hypothesis of mechanical decoupling, since vibrational relaxation on a shorter timescale than electronic de-excitation would result in a suppression of the HL peaks beyond the simplified Franck-Condon model presented above. Therefore, we conclude that the mechanical decoupling implied by the molecular design of the foot structure increases the lifetime of vibrational excited states of the chromophore and thus facilitates HL.

## Discussion
The combination of tailor-made model compounds and high-end STM-based luminescence experiment provides improved light emission with high spectral resolution, intensity and stability for self-decoupled molecules, providing a route for efficient single molecular light

**Fig. 6 | Synthetic approach to tolane based tripodal platform Tol-Tpd-H.** Reaction conditions: **a** AcCl, toluene, 120 °C; **b** TMSC≡CMgBr, THF, toluene; **c** TMSC ≡ CH, Pd(PPh₃)₂Cl₂, CuI, NEt₃, 70 °C; **d** K₂CO₃, MeOH, DCM; **e** Pd(PPh₃)₄, CuI, NEt₃, 80 °C; **f** K₂CO₃, MeOH, THF.

sources. Crucial for this approach is that the foot structure provides a rigid anchor to the substrate that overcomes van der Waals attraction of the chromophore to the substrate providing upright oriented compact NDI chromophores upon spray deposition onto the sample and subsequent annealing. Further, this process offers the possibility of upscaling with the ability to orient the transition dipoles for maximizing the quantum yield and orienting push-pull complexes in the right direction within an OLED structure. It might help to overcome the limitations of OLEDs due to their disordered structure.

The single molecule junction has a very high efficiency with up to $2.7 \times 10^{-3}$ photons per electron and is robust enough to allow for detailed STML experiments, including photon maps of an individual molecule. Furthermore, the chromophore mounted on the tripodal scaffold is also mechanically decoupled, increasing the lifetime of vibrational excited states indicated by hot luminescence bands. In principle, the approach with even extended linkers could permit the study of chromophores with much longer recombination lifetimes making single molecules available as highly efficient single photon sources of narrow band width.

The reported experiments document the potential of the tripodal foot structure and we are currently exploring its potential and limitation for the spatially controlled immobilization of functional subunits, ranging from optically addressed chromophores to molecular machinery.

## Methods
### Synthetic strategy
The synthesis of an extended platform **Tol-Tpd**-H (Fig. 6) started with preparation of trityl alcohol **1** according a modified literature procedure, which is based on monolithiation of 1,4-dibromobenzene and its nucleophilic addition to diethyl carbonate[48]. Subsequent nucleophilic substitution of trityl alcohol **1** with acetyl chloride afforded the desired trityl chloride **2** in quantitative yield. While trityl chloride is moisture sensitive and decomposes, it was used in the next step without further purification. Reaction of trityl chloride **2** with freshly prepared 2-(trimethylsilyl)ethynylmagnesium bromide afforded trimethylsilyl (TMS) protected 3,3,3-tris(4-bromophenyl) propyne scaffold **3** in 78% yield, which is suitable for the further functionalization through Sonogashira cross-coupling reactions with the corresponding phenylacetylene **5**. The thiol group of 3-bromothiophenol was protected as 2-(trimethylsilyl)ethyl derivative according published procedure with trimethyl(vinyl)silane through a radical reaction in the presence of 2,2′-azobis(2-methylpropionitrile) (AIBN) as a radical initiator to provide 2-(trimethylsilyl)

ethyl (TMSE) derivative **4** in 93% yield[32]. The following Sonogashira reaction of bromo derivative **4** with trimethylsilylacetylene and the subsequent cleavage of the TMS protecting group led to the *meta*-substituted phenylacetylene **5** with high yield. This phenylacetylene was coupled further with the previously prepared 3,3,3-tris(4-bromophenyl)propyne scaffold **3** via Sonogashira protocol to afford an extended platform **Tol-Tpd**-TMS in 77% yield. Final deprotection of TMS group yield the desired extended platform **Tol-Tpd**-H in 97%, which can be modularly functionalized in the perpendicular position.

Final assembly of tripodal chromophores **Tol-Tpd-sNDI**-Ac and **Tol-Tpd-nNDI**-Ac is outlined in Fig. 7. Under Sonogashira conditions, we mounted previously prepared **sNDI**-I derivative[23] on the tolane based tripodal platform **Tol-Tpd**-H in 74% yield. Due to the poor solubility of NDI derivatives in neat amines, THF was used as a co-solvent. The subsequent transprotection of TMSE protected thiolate **Tol-Tpd-sNDI**-TMSE afforded the desired target molecule **Tol-Tpd-sNDI**-Ac in 62% yield. Similar to compound **Tol-Tpd-sNDI**-TMSE, we mounted **nNDI**-I derivative[23] on the tolane based tripodal platform **Tol-Tpd**-H in 71% yield. The transprotection of **Tol-Tpd-nNDI**-TMSE to the corresponding thioacetate afforded the target molecule **Tol-Tpd-nNDI**-Ac in 76% yield. Both tripodal chromophores **Tol-Tpd-sNDI**-Ac and **Tol-Tpd-nNDI**-Ac bearing three thiacetate anchoring groups at *meta*-positions relative to the ethynyl groups, which are easily cleaved in situ during their deposition, provided successful coupling to the metal surfaces via three sulfur bonds.

### General experimental details and materials
All starting materials and reagents were purchased from commercial suppliers Alfa Aesar (Karlsruhe, Germany), Sigma-Aldrich (Schnelldorf, Germany), TCI Chemicals Europe (Zwijndrecht, Belgium), Merck (Darmstadt, Germany) and used without further purification. The purity of all commercially available chemicals used was higher than 98%. Solvents utilized for crystallization, chromatography and extraction were used in technical grade. Anhydrous tetrahydrofuran and dichloromethane were taken from MBraun Solvent Purification System equipped with drying columns. Triethylamine was dried and distilled from CaH₂ under nitrogen atmosphere. TLC was performed on silica gel 60 F254 plates, spots were detected by fluorescence quenching under UV light at 254 and 366 nm. Column chromatography was performed on silica gel 60 (particle size 0.040−0.063 mm). Compound **4**[32], **sNDI**-I, and **nNDI**-I[23] were prepared according to a published procedure. Detailed synthetic procedures for all previously unpublished compounds and their full characterization are provided in Supplementary Information. All compounds were purified by

**Fig. 7 | Synthetic approach towards tripodal chromophores.** Reaction conditions: **a** CuI, Pd(PPh$_3$)$_4$, NEt$_3$, THF, 74% (**Tol-Tpd-sNDI**-TMSE), 71% (**Tol-Tpd-nNDI**-TMSE); **b** AcCl, AgBF$_4$, DCM, 62% (**Tol-Tpd-sNDI**-Ac), 76% (**Tol-Tpd-nNDI**-Ac).

chromatography and fully characterized by means of conventional NMR, FTIR, and UV–vis spectroscopy, mass spectroscopy, as well as by elemental analysis.

## Equipment and measurements

All NMR spectra were recorded on a Bruker Avance 500 spectrometer at 25 °C in CDCl$_3$. $^1$H NMR (500 MHz) spectra were referred to the solvent residual proton signal (CDCl$_3$, $\delta_H = 7.24$ ppm). $^{13}$C NMR (126 MHz) with total decoupling of protons were referred to the solvent (CDCl$_3$, $\delta_C = 77.23$ ppm). For correct assignment of both $^1$H and $^{13}$C NMR spectra, $^1$H-$^1$H COSY, $^{13}$C DEPT-135, HSQC, and HMBC experiments were performed. UV–vis absorption spectra were recorded with an Agilent Cary 5000 spectrophotometer in a 1 cm quartz cell at ambient temperature in dichloromethane. Fluorescence PL spectra were measured with a Varian Cary Eclipse Fluorescence spectrometer at room temperature in a 1 cm quartz cell in dichloromethane. The molecule **Tol-Tpd-sNDI**-Ac was excited at 370 nm and both excitation and emission resolution were set to 20 nm. The molecule **Tol-Tpd-nNDI**-Ac was excited at 366 nm and both excitation and emission resolution were set to 10 nm. EI MS spectra were recorded with a Thermo Trace 1300-ISQ GC/MS instrument (samples were dissolved in dichloromethane or introduced directly using direct injection probes DIP, DEP) and $m/z$ values are given along with their relative intensities (in %) at an ionization voltage of 70 eV. High-resolution mass spectra were recorded with a Bruker Daltonics (ESI microTOF-QII) mass spectrometer. IR spectra were recorded with a Nicolet iS50 FTIR spectrometer under ATP mode. Analytical samples were dried at 40–100 °C under reduced pressure ($\approx 10^{-2}$ mbar). Melting points were measured with a Büchi Melting point M-560 apparatus and are uncorrected. Elemental analyses were obtained with a Vario MicroCube CHNS analyser. The values are expressed in mass percentage. The NMR spectra of all previously unpublished molecules are given in Suppl. Figs. 10-29, along with the molecular structure and atom numbering used for the full assignments of signals in the NMR spectra. The same parts of more complex molecular structures are marked with the same color code for better identification.

## STM and STML measurements

Low-temperature STM and STML measurements were performed using a homemade UHV STM with optical access operated at 4.4 K[34]. **Tol-Tpd-sNDI**-Ac and **Tol-Tpd-nNDI**-Ac molecules were deposited by using a spray deposition technique onto a Au(111) substrate[23,32] cleaned by repeated cycles of sputtering and annealing. Therefore, the clean Au(111) crystal is placed directly opposite to a pinhole in a rough vacuum chamber ($\approx 1 \times 10^{-2}$ mbar). About 1 µL of molecular solution (1 mg of molecules in 1 mL of dichloromethane) is being sucked in through the pinhole so that small droplets of the solution land on the sample[6,23,32], see Suppl. Fig. 1). Then the sample is directly transferred into the UHV chamber. Thereafter, the sample is annealed at ≈373 K to promote the on-surface cleavage of the acetyl protection groups and to facilitate the correct orientation of the complex with the sulfur anchors on the Au(111) surface. After the annealing, the sample was transferred to our custom-built low temperature UHV STM (4.4 K, ≈10$^{-10}$ mbar).

Our STM is equipped with optical access to collect the light emitted from the tunneling junction[34]. Light emitted from the junction is guided into an optical fiber (core diameter 200 µm) via a parabolic mirror microfabricated by using 3D direct laser writing. The fiber goes into a spectrometer (150 mm focal length) equipped with a grating of 300 groves mm$^{-1}$. The optical resolution of this setup is about 8 nm when the entrance slit is fully open (400 µm) and can be reduced to about 2 nm when the entrance slit is closed to 10 µm.

All photon spectra presented in the manuscript are corrected for the collection efficiency of the detector (see Suppl. Fig. 2a). The losses in the microfabricated mirror tip are unknown, so the presented figure of collection efficiency is an upper limit and, consequently, the values given for the quantum efficiency of the emitter given in the main manuscript are lower limits.

The plasmonic response of the mirror tip used in this work shows a single broad peak centered around 2 eV (see Suppl. Fig. 2b).

## Computational details

The structures of **Tol-Tpd-sNDI**-Ac and **Tol-Tpd-nNDI**-Ac were pre-optimized using the r$^2$SCAN functional[49] in combination with the def2-TZVP basis set[50]. Excitation spectra were then obtained using the TMHF ab initio local hybrid functional[51]. Energy and density thresholds were set to 10$^{-8}$ and 10$^{-6}$ a.u. respectively. A grid of size "veryfine" was used[52]. All DFT calculations in vacuum were carried out using TUR-BOMOLE V7.7[53].

## Reporting summary

Further information on research design is available in the Nature Portfolio Reporting Summary linked to this article.

## Data availability

The data that support the findings of this study are available from the corresponding authors upon request.

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

## Acknowledgements

This work was supported by the Helmholtz Association via the programs Natural, Artificial, and Cognitive Information Processing (NACIP) and Materials Systems Engineering (MSE). V.R. gratefully acknowledges funding by the Deutscher Akademischer Austauschdienst (DAAD) via the "Research Grant - Doctoral Programmes in Germany, 2017/18 (57299294)". N.B. acknowledges financial support by the DFG Grant (MA 2605/6-1). M.M. acknowledges support from the 111 Project (90002-18011002) and the Swiss National Science Foundation (SNF Grant no. 200020-207744). V.R. and W.W. gratefully acknowledge financial support from the Deutsche Forschungsgemeinschaft (DFG, German Research Foundation) through the Collaborative Research Center "4 f for Future" (CRC 1573, project number 471424360) project C1. We acknowledge support by the KIT-Publication Fund of the Karlsruhe Institute of Technology.

## Author contributions

N.B. and M.V. performed the organic synthesis, and carried out the chemical analysis and molecular spectroscopy. M.V. and M.M. proposed the molecular design. V.R., G.D. and L.G. performed the STM experimental studies. V.R. and L.G. analysed the STM data. C.H. performed the computational studies. M.M., W.W., L.G. and M.V. conceived and supervised the work. V.R., N.B., C.H., M.M., W.W., L.G. and M.V. co-wrote the manuscript. All authors discussed the results and commented on the manuscript.

## Funding

## Competing interests

The authors declare no competing interests.
