## [Peer Review File · Nature Communications]

Hot luminescence from molecular chromophores electrically and mechanically self-decoupled by tripodal scaffoldsREVIEWER COMMENTS

Reviewers #1 and #2, who co-reviewed your work (Remarks to the Author):

In their manuscript, Rai et al. report the successful synthesis of chromophores with tripodal anchoring groups that, upon deposition onto a Au(111) surface, lead to an upright configuration and decoupling of the chromophore from the substrate. STML on these molecules shows not only that the embedded chromophores are indeed sufficiently decoupled to prevent luminescence quenching, but also that anti-Kasha emission from the vibrationally excited S₁ to the S₀ ground state is promoted.

While this is not the first attempt to design molecules that provide this kind of decoupling by their chemical design, this is the first study that shows a reproducible orientation of the molecules upon adsorption and thereby also reproducible STML spectra. In addition, the presented anchoring platform and linker group provide excellent decoupling, resulting in bright electroluminescence. The results are beautiful, especially considering that this kind of upright-standing configuration is certainly not the easiest to handle from an experimental point of view, nicely presented, and generally well discussed. However, the conclusions drawn from the observation of anti-Kasha emission are missing some further discussion, as commented below:

1. p. 9, last quarter of the paragraph: The authors state that anti-Kasha emission has not been observed in STML of molecules lying flat on an insulating layer. This is not correct: The violation of Kasha's rule has been observed in STML on PTCDA (Nature 570, 210 (2019) Figs. 2 and 3 in the main text and Figs. 3 and 4 in the extended data). Also the Q_y luminescence observed on H₂Pc is, although it does not stem from vibrationally excited states, strictly speaking anti-Kasha emission. The statement should be corrected and clarified accordingly.

2. The authors state on p. 3 at the end of the second paragraph that the observation of hot luminescence is related to the mechanical decoupling and the resulting increase in lifetime of the vibrational modes. This is a strong statement that seems, as of now, not fully supported. In order to compare the lifetimes, comparable STML measurements on the used chromophores adsorbed on a 'common' decoupling layer would be necessary. Could the fact that the authors observe anti-Kasha emission simply be related to the Franck-Condon factors of the corresponding transitions within the used chromophores that potentially favor the observed hot luminescence as compared to for example in ZnPc? In addition, the intensity of the hot luminescence peaks is very low. This in combination with the generally very high collection efficiency in the used setup (compared to other setups) might allow the observation of such unlikely decay paths in the first place. Can this be ruled out (for example by comparing the relative intensities of the LE and HE vibronic transitions of the 0-0 line with previously published work on other molecules)? In the absence of stronger evidence linking the presence of HE peaks and vertical decoupling from the substrate, we would suggest to soften the claim.

In addition, we have some more general comments, as detailed in the following:

3. p. 2, 2nd paragraph: The authors describe the adsorption properties of different molecular subunits (binding energy, adsorption pattern, etc.). It would be nice to add references, especially for the binding energies.

4. p. 3 and p. 7, end of respective 1st paragraph: Plasmonic enhancement of transitions with a transition dipole moment perpendicular to the surface (i.e., parallel to the tip-axis) has been shown before outside the STM-community (e.g. Appl. Phys. Lett. 85, 3863–3865 (2004)). Here, it would be nice to add some of the earlier literature on this effect, too.

5. p. 3, 2nd paragraph: The authors highlight the spatial resolution in STML experiments. The spatial resolution that is presented in this work is very comparable to experiments performed on for example suspended molecules (showing the molecules as broad protrusions with no 'internal features').

Considering the geometry of the molecule this is not surprising, however, highlighting the spatial resolution in this context might be a bit too much. It would be nice if this statement was mitigated.

6. Fig. 3: Is the photon map in Fig. 3b normalized by the current? Do the authors also have constant-current photon maps? Considering the non-planarity of the molecule this might provide some additional information especially at the outer part of the molecule, where in constant-height the tip is very far away from the molecule/chromophore.

7. In sNDI, the vibronic transitions are very intense with respect to the 0-0 line, especially compared

to nNDI. Do the authors know why that is?

8. Could the authors identify/reach the ionic resonances of the molecules? In general, dI/dV spectra would be highly appreciated, especially when it comes to discussing STML mechanisms. It would be interesting to see whether and how resonant tunneling influences the electroluminescence, since the presented data suggests inelastic energy transfer as the dominating excitation mechanism within the used bias voltage range.

Considering the above-mentioned points, we recommend the article for publication in Nature Communications after the minor revisions pointed out above.

Reviewer #3 (Remarks to the Author):

The manuscript deals with an experimental system consisting of self-decoupled tripodal chromophores adsorbed on a metal surface. In this configuration, the emitter parts of the molecules are oriented perpendicular to the surface, which is advantageous for a more efficient light emission in a STML measurement. Electroluminescence is studied and provides well-resolved and relatively intense photon emission spectra containing sharp vibronic spectral features, including hot vibrational bands. The authors infer from this observation that the Kasha's rule is violated and explain it by mechanical and electronic decoupling of the molecules from the substrate, which should lead to an increased lifetime of the molecular vibrations.

The paper is well written and contains a large volume of details about the chemical processes leading to the synthesis of the tripodal chromophores. The analysis of the differing quantum efficiency of the photon-emission process between the two chemical species (Tol-Tpd-sNDI and Tol-Tpd-nNDI) is interesting and revealing, although the idea of using STML to investigate emitters decoupled from the surface by tripodal supports is not novel.

The discussion of electroluminescence spectra, however, is, in my opinion, lacking more in-depth analysis of the available results, in particular:

1) The vibronic features in the spectra are clear and very well resolved, but not sufficiently discussed. What are the vibrational modes involved? Can we learn from the spectra something about their localization/mechanical decoupling? What are the vibrational populations of the modes featuring hot luminescence (e.g. estimated from the ratios of the vibronic peak intensities)? Can we say something more about the vibrational decay rates, pumping dynamics?

2) The authors outline the physical mechanisms involved in the formation of the photon maps (Fig. 3), but do not provide any theoretical simulation (which has become standard nowadays). Instead, only spatial orientation of transition dipoles is discussed in the SI. Why don't the authors calculate the maps using the outputs of the TDDFT calculations (extracting the transition densities, not just NTOs) following some of the former works, e.g. [Phys. Rev. X 12 011012 (2022), ACS Nano 16, 1082 (2022)] or in a simplified form as done in [Phys. Rev. Lett. 130, 126202 (2023)]?

Also, I have concerns regarding certain strong statements and argumentations in the manuscript:

3) Page 8 bottom: "...is an indication that the chromophore is excited by energy transfer via gap plasmons". Although the authors are right that this mechanism has been discussed before, it is not the only possible interpretation, when the tip is in tunnelling contact with the molecule. Besides, the current theoretical understanding is far from being complete. In this case other previously discussed mechanisms can dominate or significantly contribute, including inelastic cotunneling. For a more credible assessment of the process leading to excitation, more data is needed, e.g. at least a dI/dV curve which will show the presence or absence of any features around the EL threshold. The EL yield

dependence on the tunnelling current at constant bias can not be solely considered as a reliable indicator because of the variable geometry of the nanocavity.

4) Page 9, discussion of hot luminescence: "..., but has not been observed in the typical STML experiments of single molecules adsorbed on thin insulating layers". See [ACS Nano 16, 1082 (2022)], Figures 1 and 4.

5) In the conclusions, the authors talk about "unprecedented spectral resolution". From the spectra the width of the peaks shown in the manuscript (Fig.4) seems to be at least several units of meV. Is the peak width limited by the intrinsic broadening, nanocavity effects or by the instrument resolution? Recent experimental studies provide EL peaks with FWHM below 0.7 meV, which is near to their respective instrumental resolution, e.g. [Science, 379(6636), 1049-1054 (2023)] or [Nat. Commun., 13, 6008 (2022)]. There are also other works claiming unprecedented resolution in the microelectronvolt range using resonant spectroscopy approaches, e.g. [Science 373, 95-98 (2021)]

Final remarks and questions to consider:

6) Did the authors observe any signatures of the charge-transfer exciton in the spectra?

7) Notation used to discuss the vibronic spectrum: The authors use the notation $S_i \nu_m$ to indicate the m -th number state of a vibrational mode ν in the electronic state S_i . However, the notation may be a bit misleading and ν_m can also be understood as the vibrational frequency of mode m . I would suggest changing the notation to make this distinction clearer.

8) On the Page 9, discussion of the hot bands "..., these are not anti-Stokes Raman lines." It is not clear why the authors mention the Raman process in this discussion, when the discussed transition is electronic.

9) In the Supplementary Information, Fig. S8, the caption says "Calculated emission spectrum", but the graph is showing "absorption [a.u.]".

10) First and second paragraph of the results and discussions have excessive details on the chemical synthesis of the molecules and additional description of the preparation procedure, this would benefit from being streamlined, best into one paragraph.

Overall, I think that the paper deserves publication. Nevertheless, I think that it can be substantially improved by providing further analysis and discussion of physics (as well as addressing all the concerns and remarks above).

Reviewer #4 (Remarks to the Author):

The manuscript by Rai et al. demonstrates the STM induced electroluminescence on self-decoupled naphthalene dimide chromophores at the single-molecule level. Such a decoupling is obtained through well-designed extended tripodal scaffolds, which not only enables efficient electronic decoupling of the chromophores from the metal substrate but also ensures the upright configuration of the mounted chromophore. Such a molecular structure is quite rigid, which further allows the photon imaging over a single chromophore molecule. The authors also observe evident hot-luminescence from a single Tol-Tpd-nNDI molecule, which is proposed to originate from a mechanical decoupling mechanism that increases the lifetimes of vibrational excited states.

The control over the electrical contact to an individual molecular emitter is very important in molecular

optoelectronics. In particular, how to design self-decoupled chromophores with structural rigidity and integrity is crucial for the electroluminescence studies on functional optoelectronic molecules. Based on their previous reports (Rai, et al., Phys. Rev. Lett. 130, 036201 (2023)), the authors further optimize and synthesize extended tripodal footing structures, which enables the realization of the well-defined electroluminescence from single Tol-Tpd-nNDI and Tol-TpdsNDI molecules. The chemical design of the self-decoupled molecular structure demonstrated in this manuscript is quite interesting. However, the physical picture for the electroluminescence is not clear. The following comments should be addressed. Figure 2 shows the electroluminescence spectra from single Tol-Tpd-nNDI and Tol-TpdsNDI molecules which are obtained at the center areas. The authors can measure and analyze more electroluminescence spectra around the molecule, which, together with the spectra acquired at the center areas, can offer valuable information on the dipole orientation as well as the field-molecule interactions.

1) Figure 4 shows the hot-luminescence peaks from a single Tol-Tpd-nNDI molecule only for four kinds of vibrational modes (namely, 191 cm^{-1} , 722 cm^{-1} , 1114 cm^{-1} , and 1658 cm^{-1}). Why should not the hot-luminescence of other vibrational mode be observed? What is the reason for the appearance of the hot-luminescence? Symmetry of the modes?

2) Mechanical decoupling is proposed to explain the observation hot-luminescence phenomena based on the arguments that hot-luminescence peaks are reported from multi-molecular layers and suspended molecular wires, rather than isolated molecules on insulating layers. Such an argument is quite weak. Can the authors explain that what is the physical picture of the mechanical decoupling? Can the authors provide some experimental and theoretical evidences to support that the mechanical decoupling indeed increases the lifetime of vibrational excited states of the chromophore?

3) The molecular electroluminescence is attributed to the excitation of inelastic tunneling electrons by energy transfer via gap plasmons. dI/dV measurement over a single Tol-Tpd-nNDI molecule is needed to strengthen the conclusion.

4) In Fig. S4, a plasmon onset energy of 1.95 eV is evidently observed when the bias voltage is set to 1.94 V. Such an observation clearly violets the energy conservation law that the maximum plasmon energy is limited by the excitation bias voltage. Can the authors explain this unexpected phenomena?

Reviewer #5 (Remarks to the Author):

The manuscript by Rai and coworkers reports on a successful design, synthesis, and study of self-decoupled chromophores that maintain their optical properties when deposited on a metallic surface. The main achievement of the authors is that their molecular system is robust and optical spectra acquired by exciting the molecule via the electrons tunneling from the STM tip match those recorded in solution and are in reasonable agreement with the values obtained from theory. This is an improvement compared to other studies demonstrating self-decoupling in STM-based optical experiments (ref. 20 and 21), where clear identification of the individual molecules was challenging (ref. 20), or the optical properties differed from molecule to molecule (ref. 21). This study also develops from earlier works of the authors where similar structures were not found to absorb in the "standing" configuration (ref. 23, 24). However, since these systems already exhibited some degree of decoupling, even though less controlled and not so robust, I struggle to identify this manuscript as a major advancement. In addition, in terms of optical characterization with the STM, no novel physical/chemical effect was observed; therefore, judging the manuscript's impact becomes difficult.

The paper is well-written, the descriptions are clear, and the scientific conclusions are sound. Overall, I am not convinced this is a good fit for Nat. Commun., but the work is clearly solid and of interest, especially for the development of novel molecular decoupling strategies. Below, I list some minor concerns and suggest improvements:

1. Fig. 4 shows the vibronic features in the emission of the nNDI structure; could the authors identify the vibrational modes associated with these transitions?

2. Related to the previous point, how is the transition energy defined? Is it the onset or the central position of the peak?
3. Fig. S3 shows a comparison between different nNDI structures. A similar comparison for the sNDI should be provided as well.
4. What are the widths of the spectral features observed in this work?
5. The light emission map in Fig. 3b is not normalized by the current. Can one learn something about the system when such an operation is performed?
6. The authors argue that the excitation mechanism relies on the inelastic tunneling process and show the tunneling current dependence at one bias voltage. Does it hold for other bias voltages, especially for the negative values? Furthermore, the authors should provide some scanning tunneling spectroscopy measurements of the system. Some discussion about the electronic decoupling of the system would be valuable for this work.
7. Fig. S6 misses labeling of the structures.
8. Fig. S8 shows a prediction of a low-lying S1 state of the sNDI, was it observed in the experiment?

Point-by-point response to the reviewers' comments:

We would like to thank all the reviewers for their detailed and careful evaluation of our manuscript entitled as "*Electrically and mechanically decoupled single chromophores by tripodal scaffolds*", followed by important and constructive suggestions for its further improvement. With the help of their comments, we were able to significantly strengthen our claims with additional analysis and by providing further experimental and theoretical data. We address all of the suggestions and comments point-by-point in our response below and indicate all of the changes made to the manuscript. In the text below, the reviewers' comments are set in italic font, while our answers, in blue, begin with the keyword "Reply:". The changes made in the manuscript are then written in red under the individual answers. In addition, all changes made to the manuscript and supplementary information are highlighted in yellow.

Reviewers 1 and 2, who co-reviewed our work:

In their manuscript, Rai et al. report the successful synthesis of chromophores with tripodal anchoring groups that, upon deposition onto a Au(111) surface, lead to an upright configuration and decoupling of the chromophore from the substrate. STML on these molecules shows not only that the embedded chromophores are indeed sufficiently decoupled to prevent luminescence quenching, but also that anti-Kasha emission from the vibrationally excited S1 to the S0 ground state is promoted. While this is not the first attempt to design molecules that provide this kind of decoupling by their chemical design, this is the first study that shows a reproducible orientation of the molecules upon adsorption and thereby also reproducible STML spectra. In addition, the presented anchoring platform and linker group provide excellent decoupling, resulting in bright electroluminescence. The results are beautiful, especially considering that this kind of upright-standing configuration is certainly not the easiest to handle from an experimental point of view, nicely presented, and generally well discussed. However, the conclusions

drawn from the observation of anti-Kasha emission are missing some further discussion, as commented below:

Reply: We would like to thank both the referees for recognizing and appreciating the importance of our contribution and their recommendations for further consideration. We hope to clarify any remaining ambiguities in the following text.

Comment-1.1 p. 9, last quarter of the paragraph: *The authors state that anti-Kasha emission has not been observed in STML of molecules lying flat on an insulating layer. This is not correct: The violation of Kasha's rule has been observed in STML on PTCDA (Nature 570, 210 (2019) Figs. 2 and 3 in the main text and Figs. 3 and 4 in the extended data). Also, the Q_y luminescence observed on H_2Pc is, although it does not stem from vibrationally excited states, strictly speaking anti-Kasha emission. The statement should be corrected and clarified accordingly.*

Reply: We thank the referees for pointing out this important paper, where light emission from a flat lying PTCDA molecules on NaCl/Au(111) has been studied. Indeed, hot luminescence peaks can be seen in the spectra. They were, however, not discussed in the main text of *Nature 570, 210 (2019)* article, but were mentioned in the extended Figure 4 figure caption, which simply escaped our attention ("In both the fluorescence and phosphorescence spectra, a small peak (labelled with *) was observed 30 meV higher in energy than the 0–0 transition peak. This luminescence peak was attributed to hot luminescence, which is defined as a transition from the vibrational excited states of S1 (or T1) to the vibrational ground states of S0").

In the case of H_2Pc , the Q_y line is due to a small splitting of otherwise degenerate orbitals and, as the referee correctly stated, has nothing to do with vibrational modes. Thus, we try to be more precise in the language, when speaking of vibrational insulation and *Kasha's* rule. In the context of our work, *Kasha's* rule would imply vibrational de-excitation of the electronically excited state before emission of a photon.

We want to emphasize that the focus of our manuscript lies in the detailed study of electroluminescence from selectively excited individual vibrational levels of a self-

decoupled molecular complex. Here, we clearly go beyond previous studies of hot luminescence (HL) in STML. In response to the helpful comments of the referees, we significantly extended our analysis of the observed HL, which further supports our claim of decoupled vibrational modes. See also reply to comment-3.4.

In the revised manuscript, we change the corresponding sentence mentioned by the referee and also clarify that we focus on HL with respect to vibrons and not on anti-Kasha's in general (Q_y line of H_2Pc , as correctly pointed out by the referee). The revised text now reads:

HL has been observed in previous STML experiments by Dong et al.³⁹ in molecular multilayers, by Chong et al.⁴¹ in molecular wires suspended between a metal surface and the STM tip and perylene-3,4,9,10-tetracarboxylic dianhydride (PTCDA) molecules on NaCl^{46,47}, but has not been observed in the typical STML experiments of single molecules adsorbed on thin insulating layers^{8,9,11,20}.

Comment-1.2 *The authors state on p. 3 at the end of the second paragraph that the observation of hot luminescence is related to the mechanical decoupling and the resulting increase in lifetime of the vibrational modes. This is a strong statement that seems, as of now, not fully supported. In order to compare the lifetimes, comparable STML measurements on the used chromophores adsorbed on a 'common' decoupling layer would be necessary. Could the fact that the authors observe anti-Kasha emission simply be related to the Franck-Condon factors of the corresponding transitions within the used chromophores that potentially favor the observed hot luminescence as compared to for example in ZnPc? In addition, the intensity of the hot luminescence peaks is very low. This in combination with the generally very high collection efficiency in the used setup (compared to other setups) might allow the observation of such unlikely decay paths in the first place. Can this be ruled out (for example by comparing the relative intensities of the LE and HE vibronic transitions of the 0-0 line with previously published work on other molecules)? In the absence of stronger evidence linking the presence of HE peaks and vertical decoupling from the substrate, we would suggest to soften the claim. In addition,*

we have some more general comments, as detailed in the following:

Reply: We thank the referee for these comments which helped us to analyse our experimental data in more detail. HL means that the times of the photonic de-excitation is similar or shorter compared to vibronic de-excitation, which is typically not the case (*Kasha's rule*). There are two possible scenarios: 1) The photonic de-excitation is drastically quickened due to coupling to gap plasmons-polaritons. 2): The vibronic de-excitation is suppressed due to mechanical decoupling. Scenario-1, which has been used to rationalize previous observations of HL, is an effect of low energy selectivity due to the broad gap-plasmon spectrum. It thus acts on all transitions in that energy range. Our plasmon spectra is rather broad and smooth in the energy range of interest (see Fig. S2). As has been pointed out by referee 4 in comment-4.1, four vibrational peaks on the LE side have a counterpart in the HL, but the LE-peak at 480 cm^{-1} clearly does not, in spite of a better match to the plasmonic spectrum. This excludes the possibility of a cavity enhanced photon emission rate or the high sensitivity of our setup as the explanation for the HL peaks.

In the revised manuscript, we focus on the nature of the different vibrational modes in the framework of their lifetimes following Scenario-2. In particular, we can clearly identify the well-known and characteristic line of the C=O imide modes in the range of 1600 cm^{-1} - 1650 cm^{-1} , which show up in the HL, while the line at around 480 cm^{-1} can be related to extended modes (Gerüstschwingungen) that also involve the foot structure of the molecular complex and does not show up in the HL. This is in agreement with Scenario-2. In order to quantify this selective behavior, we estimated the experimental Franck-Condon factors from the data. The main line is related to a transition without change of the vibrational quantum number in two steps from the electronic ground state to the electronically excited state and back (i.e. $0 \rightarrow 0$ and $0 \rightarrow 0$ vibrational modes). On the LE side, the excitation is from vibrational states $0 \rightarrow 0$ followed by a de-excitation in form of $0 \rightarrow 1$, i.e. an energy loss of one (or more) vibrational quantum. Thus, the ratio of the intensities for both lines is given the ratio of the corresponding Franck-Condon factors

0- >0/0->1. For the HL, we get the Franck-Condon factors 0->1 and 1->0. In lowest order, this would result in a ratio of the LE and HE lines again of the Franck-Condon factors 0- >0/0->1. We tested for this identity and show that the two ratios as function of voltage and current for the C=O mode of the chromophore are indeed very similar (see Fig. 5). This leaves no open question about the vibrational decoupling, as vibrational relaxation on the time scale of the electronic de-excitation would result in a deviation from this ratio. In contrast to this, the low energy and de-localized Gerüstschwingung can couple to the substrate and the HL peak is absent, i.e. the vibronically excited state is de-excited, before the photon is emitted (*Kasha's rule*). We hope with this additional quantitative analysis, the skepticism of the referee is taken care of.

We note that the intensity of the LE peaks in comparison to the main peak is not particularly high but has a typical value also seen in other studies. It is about 1/10. Our analysis suggests that if the HE peaks are due to the Franck-Condon factors, they should be of the order of $(1/10)^2$, i.e. they should be very weak compared to the main peak.

This analysis is now presented in the main text.

Comment-1.3 p. 2, 2nd paragraph: *The authors describe the adsorption properties of different molecular subunits (binding energy, adsorption pattern, etc.). It would be nice to add references, especially for the binding energies.*

Reply: We follow the reviewer's suggestion and add a few citations to the reviews, in which all details are available or cited. However, we decided not to report the values of the binding energies of physisorption and chemisorption because the values given previously correspond to simple organic molecules, not to the complex molecules that are the subject of this publication.

Comment-1.4 p. 3 and p. 7, end of respective 1st paragraph: *Plasmonic enhancement of transitions with a transition dipole moment perpendicular to the surface (i.e., parallel to the tip- axis) has been shown before outside the STM-community (e.g. Appl. Phys. Lett. 85, 3863–3865 (2004)). Here, it would be nice to add some of the earlier literature on this effect, too.*

Reply: We agree with the referee and follow the suggestion to add the citations.

Comment-1.5 p. 3, 2nd paragraph: *The authors highlight the spatial resolution in STML experiments. The spatial resolution that is presented in this work is very comparable to experiments performed on for example suspended molecules (showing the molecules as broad protrusions with no ‘internal features’). Considering the geometry of the molecule this is not surprising, however, highlighting the spatial resolution in this context might be a bit too much. It would be nice if this statement was mitigated.*

Reply: The referee is right that in agreement with previous work on self-decoupled molecules, our STM topography is "showing the molecules as broad protrusions with no ‘internal features’". Here, however, we intended to illustrate, that in our experiments we can actually scan the molecule with the tip without touching or displacing it and can thus laterally resolve the light emission as function of the position of current injection. So far, spatially resolved light emission has not been reported for self-decoupled molecules and presents a major step forward. We now include both, constant height and constant current photon maps of the single upright nNDI molecule, as also requested by referee 3 (comment-3.2) and the next comment. In the corresponding text, we make clear that we do not observe sub-molecular features, which is well expected for an upright molecule, as also mentioned by the referee and in full agreement with the simulated photon map that we added to our revised manuscript in response to comment-3.2.

We modified the wording on page 3 to make this point clear. The modified sentence now reads: **The integrity and stability of the immobilized molecule enables systematic STM-induced luminescence (STML) experiments including photon maps, which so far were only possible for molecules decoupled via insulating layers.**

Comment-1.6 Fig. 3: *Is the photon map in Fig. 3b normalized by the current? Do the authors also have constant-current photon maps? Considering the non-planarity of the molecule this might provide some additional information especially at the outer part of the molecule, where in constant- height the tip is very far away from the molecule/chromophore.*

Reply: Figure 3b was recorded in constant height mode. Division of the photon counts by the current diverges when going away from the center of the molecule, as the current drops essentially to zero. Normalization from constant height maps thus is not helpful. Instead, constant current photon maps give better information. We thus now also present this data in the revised manuscript (see Fig. 4 a-e). Indeed, due to the very sharp structure of the upright molecule, no internal features can be revealed by the maps.

Comment-1.7 *In sNDI, the vibronic transitions are very intense with respect to the 0-0 line, especially compared to nNDI. Do the authors know why that is?*

Reply: We thank the referee for this keen observation. The main line of sNDI sits at 2.25 eV and is thus close to the absorption edge of the Au substrate at about 2.35 eV where the intensity of the gap plasmon is reduced (see Fig. S2 in the SI). Thus, higher energetic lines (main line) are more damped than lower energy lines (LE lines). This makes a quantitative analysis of the ratios rather complex and would require a full simulation of the plasmonic properties of the junction. In contrast, the lines of interest in nNDI are all below the energy of the sNDI main line.

Comment-1.8 *Could the authors identify/reach the ionic resonances of the molecules? In general, dI/dV spectra would be highly appreciated, especially when it comes to discussing STML mechanisms. It would be interesting to see whether and how resonant tunneling influences the electroluminescence, since the presented data suggests inelastic energy transfer as the dominating excitation mechanism within the used bias voltage range.*

Reply: We thank the referee for the question. Indeed, we did measure dI/dV spectra on Tol-Tpd-nNDI and Tol-Tpd-sNDI (see Fig. 3b of the revised manuscript). There is no resonant tunneling at the onset of the main line of nNDI, excluding an inelastic co-tunneling process for excitation (see also comment-3.3). For sNDI, we cannot make a decisive statement, as the onset of luminescence is close to the peak in the differential conductance.

Considering the above-mentioned points, we recommend the article for publication in

Nature Communications after the minor revisions pointed out above.

Reviewer 3:

The manuscript deals with an experimental system consisting of self-decoupled tripodal chromophores adsorbed on a metal surface. In this configuration, the emitter parts of the molecules are oriented perpendicular to the surface, which is advantageous for a more efficient light emission in a STML measurement. Electroluminescence is studied and provides well-resolved and relatively intense photon emission spectra containing sharp vibronic spectral features, including hot vibrational bands. The authors infer from this observation that the *Kasha's* rule is violated and explain it by mechanical and electronic decoupling of the molecules from the substrate, which should lead to an increased lifetime of the molecular vibrations.

Comment-3.0: *The paper is well written and contains a large volume of details about the chemical processes leading to the synthesis of the tripodal chromophores. The analysis of the differing quantum efficiency of the photon-emission process between the two chemical species (TolTpd-sNDI and Tol-Tpd-nNDI) is interesting and revealing, although the idea of using STML to investigate emitters decoupled from the surface by tripodal supports is not novel.*

Reply: We thank the referee for his acknowledging the quality of our manuscript. The referee is right, the idea of emitters decoupled from the surface by tripodal scaffolds is not new and we cited the corresponding publications. We go beyond this initial work, by showing that the stability of the surface-anchored molecule allows systematic and spatially resolved STML experiments including HL, which so far were only possible for molecules decoupled via insulating layers. For illustration of the drastically improved reproducibility, we show the published spectra from the work that is closest to our concept (T. Ijaz et al., *Appl. Phys. Lett.* **115**, 173101 (2019)), in the left part of the figure below directly next to ours. We are convinced that the development and synthesis of a chromophore on a

Editorial Note: Fig. 3 below is reprinted from Ijaz, T., et al., Self-decoupled tetrapodal perylene molecules for luminescence studies of isolated emitters on Au(111), Appl. Phys. Lett. 115, 173101 (2019), with the permission of AIP Publishing.

tripodal scaffolds and the corresponding experimental proof of the concept present an important advance which is highly significant to the field, in full agreement with the scope of Nature Communications.

Comment-3.1 The discussion of electroluminescence spectra, however, is, in my opinion, lacking more in-depth analysis of the available results, in particular: The vibronic features in the spectra are clear and very well resolved, but not sufficiently discussed. What are the vibrational modes involved? Can we learn from the spectra something about their localization/mechanical decoupling? What are the vibrational populations of the modes featuring hot luminescence (e.g. estimated from the ratios of the vibronic peak intensities)? Can we say something more about the vibrational decay rates, pumping dynamics?

Reply: We want to thank the referee for pushing us to further analyze our data. In the revised manuscript, we discuss the nature of the different vibrational modes in more detail as is described in detail in our reply to comment-1.2. In particular, the characteristic C=O modes of the chromophore show up in the HL. The line at around 480 cm^{-1} which can be related to extended modes (Gerüstschwingungen) that also involve the foot structure of the molecular complex, does not show up in the HL. As suggested by the referee, we further analyzed the intensities of the vibrational peaks and extracted the ratios of the

Franck-Condon factors to further corroborate our claim of reduced vibrational decay rates due to mechanical decoupling.

Comment-3.2 *The authors outline the physical mechanisms involved in the formation of the photon maps (Fig. 3), but do not provide any theoretical simulation (which has become standard nowadays). Instead, only spatial orientation of transition dipoles is discussed in the SI. Why don't the authors calculate the maps using the outputs of the TDDFT calculations (extracting the transition densities, not just NTOs) following some of the former works, e.g. [Phys. Rev. X 12 011012 (2022), ACS Nano 16, 1082 (2022)] or in a simplified form as done in [Phys. Rev. Lett. 130, 126202 (2023)]?*

Reply: Now we do show the experimental constant height and constant current photon maps (see also comments 1.5 and 1.6 raised by referees 1 and 2). Also, a simulated photon map obtained from the excited-state TD-DFT results has been added. It matches the experimentally observed photon map closely. Technical details concerning the simulated photon maps have been added in the SI. We are not aware of any publication in this field, that provides lateral photon maps based on theoretical calculation involving Franck-Condon processes including LE and HE peaks.

Comment-3.3 *Also, I have concerns regarding certain strong statements and argumentations in the manuscript: Page 8 bottom: "...is an indication that the chromophore is excited by energy transfer via gap plasmons". Although the authors are right that this mechanism has been discussed before, it is not the only possible interpretation, when the tip is in tunnelling contact with the molecule. Besides, the current theoretical understanding is far from being complete. In this case other previously discussed mechanisms can dominate or significantly contribute, including inelastic cotunneling. For a more credible assessment of the process leading to excitation, more data is needed, e.g. at least a dI/dV curve which will show the presence or absence of any features around the EL threshold. The EL yield dependence on the tunnelling current at constant bias can not be solely considered as a reliable indicator because of the variable geometry of the nanocavity.* **Reply:** We thank the referee for the question. In the revised

manuscript, we show dI/dU spectra on Tol-Tpd-nNDI and Tol-Tpd-sNDI (see Fig. 3b of the revised manuscript) and in particular, we discuss the absence of any features in the dI/dU around the EL threshold in case of the nNDI molecule, excluding inelastic co-tunnelling processes. See also reply to comment-1.8 where the referee has similar question.

Comment-3.4 *Page 9, discussion of hot luminescence: "..., but has not been observed in the typical STML experiments of single molecules adsorbed on thin insulating layers". See [ACS Nano 16, 1082 (2022)], Figures 1 and 4.*

Reply: We thank the referee for pointing out that hot luminescence indeed has been reported for particular PTCDA molecules on insulating films. We have corrected this in the revised manuscript (see reply to comment 1.1) and cite the mentioned ACS Nano paper, but would like to stress, that in this paper, hot luminescence is not mentioned at all but could indeed have been deduced when looking closely at the mentioned figures. This simply escaped our attention.

Comment-3.5 *In the conclusions, the authors talk about "unprecedented spectral resolution". From the spectra the width of the peaks shown in the manuscript (Fig.4) seems to be at least several units of meV. Is the peak width limited by the intrinsic broadening, nanocavity effects or by the instrument resolution? Recent experimental studies provide EL peaks with FWHM below 0.7 meV, which is near to their respective instrumental resolution, e.g. [Science, 379(6636), 1049-1054 (2023)] or [Nat. Commun., 13, 6008 (2022)]. There are also other works claiming unprecedented resolution in the microelectronvolt range using resonant spectroscopy approaches, e.g. [Science 373, 95–98 (2021)]*

Reply: We want to thank the referee for this comment. Indeed, there have been several reports of better spectral resolution than the data we present in this work (the resolution in our STML spectra is limited by our detector to about 2 nm as described in the SI). However, the spectral resolution and reproducibility we present here is unprecedented for self-decoupled molecules. We change and specify the wording in the corresponding

sentences, which now read:

Page 8: In addition, the mechanical stability of the **Tol-Tpd** platform allows us to perform spatially resolved STML measurements, which is to the best of our knowledge unprecedented on such self-decoupled molecules and clearly shows the high stability of the molecular junction.

Page 12: The combination of tailor-made model compounds and high-end STM-based luminescence experiment provides light emission with spectral resolution, intensity and stability that are unprecedented for self-decoupled molecules, providing a route for efficient single molecular light sources.

Comment-3.6 *Did the authors observe any signatures of the charge-transfer exciton in the spectra?*

Reply: We thank the referee for this question. The main lines of nNDI and sNDI observed in our experiment match the photo-luminescence line in solution. We do not see any signatures of charge-transfer excitons in the spectral region, this work focuses on. We have indications for an additional and much weaker, higher energy line for nNDI that seems not to be caused by a charge-transfer exciton. Its origin is not fully understood at the moment and is unrelated to the main line and its LE and HE satellites. We would like to refrain from comments on the nature of that line, until we have understood it. Theoretical TD-DFT investigations using advanced local hybrid functionals, being in principle able to describe charge-transfer excitons, also do not indicate that the observed lines have charged-transfer character. However, we already noted in the SI that a charge-transfer transition is the cause for the low efficiency of Tol-Tpd-sNDI. This CT transition is at the ground-state geometry the S2 state, but the S1 state on the full hypersurface according to theoretical excited state optimizations that were performed, see especially Fig. S7.

Comment-3.7 *Notation used to discuss the vibronic spectrum: The authors use the notation $S_{i v_m}$ to indicate the m -th number state of a vibrational mode v in the electronic state S_i . However, the notation may be a bit misleading and v_m can also be understood*

as the vibrational frequency of mode m . I would suggest changing the notation to make this distinction clearer.

Reply: We thank the referee for this comment. We now explicitly define the notation in the main manuscript to make it clearer.

Comment-3.8 *On the Page 9, discussion of the hot bands "..., these are not anti-Stokes Raman lines." It is not clear why the authors mention the Raman process in this discussion, when the discussed transition is electronic.*

Reply: We thank the referee for this comment. Indeed, the process we observe is not a Raman process in the strict sense, as we do not shine light on the sample. In our case the molecule is excited only indirectly by plasmons. We thus omit "Raman" and only speak of Stokes (creation of a vibron due to the electronic transition) and anti-Stokes (annihilation of a vibron due to the electronic transition) lines. In our experiment, we can clearly exclude an anti-Stokes process. Instead, the intensities can be understood based on the Franck-Condon factors and the gain

peaks appear only at a bias voltage corresponding to their photon energy (hot luminescence).

In the revised manuscript, we now write: Note that, since the HE-band peaks are not excited together with the main Q_n line, but only when the tunnelling electrons have the energy of the corresponding transition, these are not involving annihilation of vibrons (which cause anti-Stokes lines).

Comment-3.9 *In the Supplementary Information, Fig. S8, the caption says "Calculated emission spectrum", but the graph is showing "absorption [a.u.]".*

Reply: We are sorry for this mistake. Of course, absorption and emission are linked by reciprocity and can be used in this context as synonyms. We corrected and aligned caption and graph. Both now read emission correctly.

Comment-3.10 *First and second paragraph of the results and discussions have excessive details on the chemical synthesis of the molecules and additional description of the preparation procedure, this would benefit from being streamlined, best into one*

paragraph.

Reply: We follow the reviewer's suggestion and streamline and slightly shorten the paragraph on molecular design and synthetic strategy.

Comment-3.11 Overall, I think that the paper deserves publication. Nevertheless, I think that it can be substantially improved by providing further analysis and discussion of physics (as well as addressing all the concerns and remarks above).

Reply: In response to the comments by the referees, in the revised manuscript, we show additional data (intensity of Q_n and Q_s lines as function of applied voltage, dI/dU spectra for both molecules, photon map at constant current, simulated photon map) and we have significantly extended the analysis of vibrational satellites (identification of the nature of the involved molecular modes with the help of DFT calculations, analysis of intensities of Q_n/LE and LE/HE peaks in a simplified Franck-Condon model). With this, we hope to address the improvements that the referee had in mind.

Reviewer 4 (Remarks to the Author):

The manuscript by Rai et al. demonstrates the STM induced electroluminescence on self-decoupled naphthalene dimide chromophores at the single-molecule level. Such a decoupling is obtained through well-designed extended tripodal scaffolds, which not only enables efficient electronic decoupling of the chromophores from the metal substrate but also ensures the upright configuration of the mounted chromophore. Such a molecular structure is quite rigid, which further allows the photon imaging over a single chromophore molecule. The authors also observe evident hot-luminescence from a single Tol-Tpd-nNDI molecule, which is proposed to originate from a mechanical decoupling mechanism that increases the lifetimes of vibrational excited states. The control over the electrical contact to an individual molecular emitter is very important in molecular optoelectronics. In particular, how to design self-decoupled chromophores with structural rigidity and integrity is crucial for the electroluminescence studies on functional optoelectronic molecules. Based on their previous reports (Rai, et al., Phys. Rev. Lett.

130, 036201 (2023)), the authors further optimize and synthesize extended tripodal footing structures, which enables the realization of the well-defined electroluminescence from single Tol-Tpd-nNDI and Tol-TpdsNDI molecules. The chemical design of the self-decoupled molecular structure demonstrated in this manuscript is quite interesting. However, the physical picture for the electroluminescence is not clear. The following comments should be addressed.

Comment-4.0 *Figure 2 shows the electroluminescence spectra from single Tol-Tpd-nNDI and Tol-Tpd-sNDI molecules which are obtained at the center areas. The authors can measure and analyze more electroluminescence spectra around the molecule, which, together with the spectra acquired at the center areas, can offer valuable information on the dipole orientation as well as the field-molecule interactions.*

Reply: We thank the referee for this helpful comment. In our revised manuscript, we show an additional photon map of Tol-Tpd-nNDI at constant current, which includes spectra on different positions of the molecule and next to it. Due to the very sharp structure of the upright molecule, the current is always injected into the apex of the molecule and there is a sharp transition from molecular luminescence to the weak plasmonic light emission when moving the tip away from the molecule (see Fig. 4d-f of the revised manuscript).

Comment-4.1 *Figure 4 shows the hot-luminescence peaks from a single Tol-Tpd-nNDI molecule only for four kinds of vibrational modes (namely, 191 cm^{-1} , 722 cm^{-1} , 1114 cm^{-1} , and 1658 cm^{-1}). Why should not the hot-luminescence of other vibrational mode be observed? What is the reason for the appearance of the hot-luminescence? Symmetry of the modes?*

Reply: We really want to thank the referee for pointing out that indeed one prominent vibrational mode at 480 cm^{-1} does not show a counter part in the HL. With the help of DFT calculations we were able to identify the nature of this line as composed of different Gerüstschwingungen which are localized also on the foot of the molecular complex. Lines that show HL can be explained by modes located on the chromophore part of the

molecule (see the new Supplementary Movies 1-5). This agrees very well with our explanation of mechanical decoupling as the main reason for the observation of HL and will be discussed in our revised manuscript. See also our reply to comment-1.2.

Comment-4.2 *Mechanical decoupling is proposed to explain the observation hot-luminescence phenomena based on the arguments that hot-luminescence peaks are reported from multi-molecular layers and suspended molecular wires, rather than isolated molecules on insulating layers. Such an argument is quite weak. Can the authors explain that what is the physical picture of the mechanical decoupling? Can the authors provide some experimental and theoretical evidences to support that the mechanical decoupling indeed increases the lifetime of vibrational excited states of the chromophore?*

Reply: With the help of the referee (see comment above), we are able to provide much stronger evidence to support our hypothesis of mechanical decoupling as the main reason for the observation of HL. See also our reply to comment-1.2.

Comment-4.3 *The molecular electroluminescence is attributed to the excitation of inelastic tunneling electrons by energy transfer via gap plasmons. dI/dV measurement over a single Tol- Tpd-nNDI molecule is needed to strengthen the conclusion.*

Reply: In our revised manuscript, we now show dI/dU measurements (see Fig. 3b of the revised manuscript) which indeed support our conclusion that the Tol-Tpd-nNDI molecule is inelastically excited. See our reply to comment 1.8.

Comment-4.4 *In Fig. S4, a plasmon onset energy of 1.95 eV is evidently observed when the bias voltage is set to 1.94 V. Such an observation clearly violet the energy conservation law that the maximum plasmon energy is limited by the excitation bias voltage. Can the authors explain this unexpected phenomena?*

Reply: Strictly speaking, the over bias emission of plasmons happens via the noise of the tunneling current (see seminal paper by P.-J. Peters, R. Berndt et al., Phys. Rev. Lett. 119, 066803 (2017)), such that plasmons of higher energy than the bias voltage are present in all cases. This process, is, however, a rather inefficient many electron process and not linear in current. Once the bias voltage hits the plasmon energy, the more efficient

one electron process sets in. The transition between both regimes in nNDI is rather sharp and on the order of tens of meV as illustrated in Fig. S4 of the SI. Similarly, our energy resolution in the optical regime is about 5 meV. To the best of our understanding, we do not observe violation of energy conservation.

Reviewer 5 (Remarks to the Author):

The manuscript by Rai and coworkers reports on a successful design, synthesis, and study of self-decoupled chromophores that maintain their optical properties when deposited on a metallic surface. The main achievement of the authors is that their molecular system is robust and optical spectra acquired by exciting the molecule via the electrons tunneling from the STM tip match those recorded in solution and are in reasonable agreement with the values obtained from theory. This is an improvement compared to other studies demonstrating self-decoupling in STM-based optical experiments (ref. 20 and 21), where clear identification of the individual molecules was challenging (ref. 20), or the optical properties differed from molecule to molecule (ref. 21). This study also develops from earlier works of the authors where similar structures were not found to absorb in the “standing” configuration (ref. 23, 24). However, since these systems already exhibited some degree of decoupling, even though less controlled and not so robust, I struggle to identify this manuscript as a major advancement. In addition, in terms of optical characterization with the STM, no novel physical/chemical effect was observed; therefore, judging the manuscript’s impact becomes difficult.

***Comment-5.0.** The paper is well-written, the descriptions are clear, and the scientific conclusions are sound. Overall, I am not convinced this is a good fit for Nat. Commun., but the work is clearly solid and of interest, especially for the development of novel molecular decoupling strategies. Below, I list some minor concerns and suggest improvements:*

Reply: We thank the referee for his acknowledging the quality of our manuscript. We agree that the idea of self-decoupled emitters decoupled not new and we cited the

corresponding publications. We clearly go beyond these initial work and we are convinced that our results present a significant advance, in full agreement with the scope of Nature Communications. See also our reply to comment 3.0.

Comment-5.1. *Fig. 4 shows the vibronic features in the emission of the nNDI structure; could the authors identify the vibrational modes associated with these transitions?*

Reply: We thank the referee for this comment and with the help of DFT calculations, we extended the analysis of the vibronic features and, in particular, identify the nature of the vibrational modes, as suggested (see also comments of the other referees).

Comment-5.2 *Related to the previous point, how is the transition energy defined? Is it the onset or the central position of the peak?*

Reply: Transition energies are defined by a Lorentzian fit function, i.e. they describe the central energy. To further clarify this, we now explicitly mention it in the caption of Figure 5.

Comment-5.3 *Fig. S3 shows a comparison between different nNDI structures. A similar comparison for the sNDI should be provided as well.*

Reply: As shown in Figure S3, the emission spectrum of nNDI molecules is highly reproducible and does not vary significantly. Following the reviewer's suggestion, we now provide the spectra on different molecules also for sNDI.

Comment-5.4 *What are the widths of the spectral features observed in this work?*

Reply: The width is limited by the resolution of our spectrometer (described in the SI) to about 2 nm (entry slit 10 μm) for some of the data on the bright nNDI and about 8 nm (entry slit fully opened) for the less bright sNDI and some measurements on the nNDI (e.g. photon maps).

Comment-5.5 *The light emission map in Fig. 3b is not normalized by the current. Can one learn something about the system when such an operation is performed?*

Reply: The constant height map cannot be normalized to the current, as the current drops to zero. Instead, we have added a constant current map to provide this information. Also see replies to comments 1.6 and 3.2.

Comment-5.6 *The authors argue that the excitation mechanism relies on the inelastic tunneling process and show the tunneling current dependence at one bias voltage. Does it hold for other bias voltages, especially for the negative values? Furthermore, the authors should provide some scanning tunneling spectroscopy measurements of the system. Some discussion about the electronic decoupling of the system would be valuable for this work.*

Reply: We follow the suggestion of the referee and also present tunneling spectra of the molecule. The observed peaks are rather sharp, as expected for decoupled orbitals. As we mention in Fig. S6 of the SI, our NTO analysis also shows that the tripod platform does not interfere with the optical excitation, efficiently decoupling the NDI moiety from the surface. In addition, we show the voltage dependence of the intensity of the Q_n as suggested by the referee. In particular, it shows that the Q_n and Q_s lines are absent at negative bias (see Fig. 3a of the revised manuscript).

Comment-5.7 *Fig. S6 misses labeling of the structures.*

Reply: We thank the referee for pointing this out and added the labels to the structures.

Comment-5.8 *Fig. S8 shows a prediction of a low-lying S1 state of the sNDI, was it observed in the experiment?*

Reply: As shown in the mentioned Figure S8, the S1 line is much weaker in intensity and at a relatively low energy. We indeed did not observe this line, probably because its energy is very near our cut off of our detector which is described in Fig. S2 of the SI.

REVIEWER COMMENTS

Reviewer #1 (Remarks to the Author):

The authors have well accounted for the reviewer's criticisms and comments. I now fully support the publication of this manuscript in Nature Communication.

Reviewer #2 (Remarks to the Author):

In their revised manuscript, Rai and co-workers have provided additional data and included a more thorough discussion, especially concerning the observation of hot luminescence. These changes have significantly improved the manuscript, so that I can now recommend its publication in Nature Communications.

I still have a few very minor comments, however, whether or not the authors choose to implement these comments will not impact my overall recommendation for publication.

p. 8, l. 236: I reckon the authors want to say here that the NIR appears at voltages above the onset of the main emission line. This would exclude excitation via consecutive charge-transfer from the electrodes to the molecule (at least in the voltage range below the NIR), I am not sure whether it can exclude inelastic co-tunneling processes as a source of excitation.

p. 10, l. 298: While I appreciate that the authors included references on hot luminescence in STML, the last part of the sentence seems to be missing some clarification, though. The authors write here that "HL [...] has not been observed in the typical STML experiments of single molecules adsorbed on thin insulating layers", while the last example (PTCDA on NaCl) is in fact exactly that.

p. 10, in l. 300 & 303 it should read "vibrational de-excitation" instead of vibronic.

Reviewer #3 (Remarks to the Author):

The authors have replied to the concerns and made the effort to better analyze and interpret the data. They've also corrected the criticized statements and added relevant references. Although the discussion about the level of significance of this work is still not closed, and the overall impression is that the dataset is not clear enough to draw precise conclusions about the underlying physics of excitation and emission, I am convinced it is a valuable contribution to the field and support its publication.

Reviewer #4 (Remarks to the Author):

The revised manuscript demonstrates commendable efforts by the authors in addressing the concerns raised by the reviewers, leading to substantial improvement in the quality of the paper. Nevertheless, despite these improvements, further clarifications on the following aspects are necessary before the paper can be considered for publication in Nature Communications.

1. The excitation mechanism is a crucial issue for STML experiments. Although the specific excitation mechanism can be controversial, I do expect the authors to discuss it in more detail, at least based on the existing models in the literatures.
2. Why are the NIR onset voltages for Tol-Tpd-nNDI and Tol-Tpd-sNDI molecules almost identical while the bias-dependent STML intensity shows quite different behaviors (Fig. 3)? Specifically, the

STML intensity for the former reaches its maximum value around 2.1 V, but for the latter it increases monotonically. Why?

3. It is unclear why the STML behaviors are quite different at positive and negative biases for both molecules. What are the potential roles of inelastic electron tunneling or carrier injection mechanisms for different molecules at different bias polarities? Can the authors explain why the STML intensity is almost zero for Tol-Tpd-nNDI when the bias voltage exceeds -2 V (Fig. 3)?

4. The dI/dV data is important for identifying the underlying excitation mechanism. Are the dI/dV data reproducible for different tip positions within a single molecule and for different molecular configurations? Are the STML intensity and dI/dV data shown in Fig. 3 collected on the same molecule and for the same tip position?

5. In the revised manuscript, the authors provide additional calculations to support the hypothesis of mechanical decoupling as the main reason for the observation of hot luminescence. However, it is not clear what kind of methods and basis sets are used for the calculation of the vibrational modes. For polyatomic molecules, there should be many vibrational modes and it is very common that the numerically calculated vibrational frequencies are not identical with the experimental observation. Thus, there could be several vibrational modes within a specific frequency range (e.g., around 485 cm^{-1}). To clearly identify the correspondence between the vibrational modes and the experimentally observed vibronic peaks, the vibrational-resolved emission spectrum should be calculated and compared with the experimental spectrum [see e.g., Fig. 2a in Nat. Commun. 12, 1280 (2021)]. The authors should try such calculations or at least calculate the Franck-Condon factors for the vibrational modes. Without these calculations, it is not convincing how the authors identify the five most intense vibronic peaks as the modes shown in Supplementary Movies 1–5.

6. In the revised manuscript, the authors “excludes the possibility of a cavity enhanced photon emission rate as the explanation for the HL peaks”. In my opinion, this conclusion should be treated with care. It is known that the lifetime of electronic excited states for a free-space molecule in vacuum is usually on the order of nanoseconds while the lifetime of vibrational excited states is on the order of picoseconds. Thus, even without the mechanical interaction between the molecule and the substrate, the vibrational excitation will decay much faster than the electronic excitation, making it very difficult to observe hot luminescence. The mechanical decoupling of the chromophore may increase the lifetimes of the chromophores’ vibrational modes, but this does not guarantee the occurrence of hot luminescence without the involvement of other factors (e.g., the Purcell effect that leads to the increase of the decay rate).

Reviewer #5 (Remarks to the Author):

The authors have submitted a revised manuscript, in which they provided additional experimental data and theoretical calculations significantly improving the quality and generality of the paper. In particular, the extended discussion on the electronic structure, vibronic spectroscopy, and mechanical decoupling is very valuable. The main concerns raised by the referees have been answered. Some minor issues remain and I list them below. Once they are addressed, I will be happy to recommend the manuscript for publication in Nature Communications.

1. My main concern in this version of the manuscript relates to the excitation mechanism. As of now, the authors consider only inelastic energy transfer. In the case of nNDI, this interpretation is reasonable for bias voltages between the onset of the emission and the onset of the NIR (2.36 V). Some data, however, are recorded at bias voltages higher than this onset, for example, the current dependence in Fig. S5 or the maps in Fig. 4, where one can expect that the excitation is realized by sequential charging and discharging of the molecule. While the inelastic tunneling excitation path still remains possible in principle, direct charging often is the dominating mechanism (see Phys. Rev. Lett. 122, 177401, 2019), which also is a one-electron process and the light intensity increases linearly with the current. In that respect, it is unfortunate that the bias dependence in the new Fig. 3 does not extend to voltages above the onset of NIR. That may also be the mechanism at play for the excitation

of sNDI. Overall, this point needs more discussion.

2. The authors now show different spectra recorded on sNDI. As they argue in the reply to one of the comments (1.7), the interpretation of these measurements is complex due to the plasmonic response. Have the authors normalized the spectra by the respective plasmonic responses of the used tips? Such an operation could help understand the role of the enhancement in these measurements.

3. Line 226: "Qn and Qs are absent for negative U". However, there is a signal visible for $V < -2$ V in the red curve in Fig. 3a. Is it of plasmonic origin? The authors should comment.

4. Lines 277-279: This sentence is not clear language-wise and should be revised.

5. Lines 291-293: I am not sure I understand this sentence, that is, isn't a vibron annihilated during emission, say from S1v1 to S0v0?

6. Lines 295-299. I found this sentence contradicting, as one would consider experiments with PTCDA on NaCl typical experiments of a single molecule adsorbed on thin insulating layers. Also, reference 20 is a paper considering a self-decoupled system.

7. Related to the same paragraph, it would be helpful for the readers to explicitly clarify the difference between the vibrational hot luminescence and emission from higher-lying excited states violating Kasha's rule. Besides the emission of H2Pc mentioned in comment 1.1, the emission from a much higher-lying state was reported in the case of porphyrins (Appl. Phys. Lett. 117, 243301, 2020).

8. The supplementary movies should be labeled with the illustrated modes.

Point-by-point response to the reviewers' comments:

We would like to thank all the reviewers for their detailed and careful evaluation of our revised manuscript entitled as "*Electrically and mechanically decoupled single chromophores by tripodal scaffolds*", and our replies to their questions. We hereby resubmit our revised version, where we address all the remaining suggestions and comments point-by-point in our response below and indicate all the changes made to the manuscript. In the text below, the reviewers' comments are set in italic font, while our answers, in blue, begin with the keyword "Reply:". The changes made in the revised manuscript are then written in red under the individual answers. In addition, all changes made to this second revised manuscript and supplementary information are highlighted in green.

Reviewers 1, 2 and 3:

We thank all three reviewers for supporting the publication. No further changes or questions are required from reviewer 1 and 3.

Reviewer 2 (Remarks to the Author):

I still have a few very minor comments, however, whether or not the authors choose to implement these comments will not impact my overall recommendation for publication.

Comment-2.1 p. 8, l. 236: *I reckon the authors want to say here that the NIR appears at voltages above the onset of the main emission line. This would exclude excitation via consecutive charge-transfer from the electrodes to the molecule (at least in the voltage range below the NIR), I am not sure whether it can exclude inelastic co-tunneling processes as a source of excitation.*

Reply: We thank the referee for pointing out that mistake. The referee is right that

the position of the NIR above the energy of the main emission line excludes excitation via consecutive charge transfer, i.e. with an intermediate state that is actually populated, while inelastic processes setting in at the energy of the main line cannot be excluded. We correct the corresponding sentence in the manuscript which now reads: "For **Tol-Tpd-nNDI**, the NIR appears clearly above the energy of the main emission line Q_n , excluding excitation via sequential charging and discharging of the molecule." We hope that this correction will avoid misunderstandings. Regarding the possible excitation mechanism, see also our reply to Comment 4.1.

Comment-2.2 p. 10, l. 298: *While I appreciate that the authors included references on hot luminescence in STML, the last part of the sentence seems to be missing some clarification, though. The authors write here that "HL [...] has not been observed in the typical STML experiments of single molecules adsorbed on thin insulating layers", while the last example (PTCDA on NaCl) is in fact exactly that.*

Reply: We now clarify the text as follows; "... has not been observed in the typical STML experiments with single molecules adsorbed on thin insulating layers, except in the case of PTCDA on NaCl."

Comment-2.3 p. 10, in l. 300 and 303 it should read "vibrational de-excitation" instead of vibronic.

Reply: The sentences were corrected as suggested.

Reviewer 4 (Remarks to the Author):

The revised manuscript demonstrates commendable efforts by the authors in addressing the concerns raised by the reviewers, leading to substantial improvement in the quality of the paper. Nevertheless, despite these improvements, further clarifications on the following aspects are necessary before the paper can be considered for publication in Nature Communications.

Comment-4.1 *The excitation mechanism is a crucial issue for STML experiments. Although the specific excitation mechanism can be controversial, I do expect the authors to discuss it in more detail, at least based on the existing models in the literatures.*

Reply: The specific excitation mechanism is not the main scope of this paper, but we follow the reviewer's suggestion and discuss the possible mechanisms in more detail: While we have clear indications for an inelastic energy transfer in the plasmonic junction, we do not see the typical signs of a carrier injection mechanism (also see the next comment and Comment 5.1). We now show voltage dependent intensity of light emission from Tol-Tpd-nNDI up to 3 V and extended the discussion of possible mechanisms in the revised manuscript which now reads (p. 8):

"Based on previous studies, this onset of STML, once the corresponding excitation energy is provided by the tunnelling electrons, is an indication that the chromophore is excited by inelastic energy transfer in the plasmonic junction. An excitation via sequential charging and discharging of the molecule would result in a step-wise increase of the photon emission rate at a higher voltage¹², which we do not observe for voltages up to 3 V (for **Tol-Tpd-nNDI** see Fig. S4). This is in full agreement with STML experiments at different tunnelling currents (see Supplementary Fig. S5)"

"For **Tol-Tpd-nNDI**, the NIR appears clearly above the energy of the main emission line Q_n , excluding excitation via sequential charging and discharging of the molecule."

We agree with the reviewer (also see Comment 2.1), that our wording was not precise enough. Light emission sets in at lower biases than the NIR, i.e. the process is not based on charging the molecule for real. The process is inelastic, and the carrier interactions in the molecule and the tunneling electrons are all purely electromagnetic and involve the electromagnetic field in the plasmonic junction. We are now more specific and use the term "inelastic energy transfer in the plasmonic junction" in accordance with recent literature and the interaction picture of quantum field theory.

Comment-4.2 *Why are the NIR onset voltages for Tol-Tpd-nNDI and Tol-Tpd-sNDI molecules almost identical while the bias-dependent STML intensity shows quite different*

behaviors (Fig. 3)? Specifically, the STML intensity for the former reaches its maximum value around 2.1 V, but for the latter it increases monotonically. Why?

Reply: The position of the NIR is determined by the charging energy of the molecule (n electrons to n+1 electrons) including the work function and the image charges. The two molecules differ only slightly and it is not surprising, that the NIR does not differ much. The optical gap is given by the energy difference of two states with the same number of electrons (n in this case). Light emission thus depends on the optical gap, and NIR on the charging energy. The onset of light emission is observed in both cases at the photon energy, which is different in the two due to the different optical gap. At higher biases, the intensity first increases, followed by a maximum and then a decrease. We can only apply about 3 V in the junction without destroying the molecules. Thus, for sNDI with a larger optical gap, we observed only an increase and perhaps a sign of saturation. We cannot apply a high enough bias to observe a decrease as well. For nNDI, the full voltage dependence was observed while not exceeding the maximal 3 V bias. We note that we have added a supplementary figure for this behaviour in nNDI. We also modified the text in the manuscript discussing this (see the previous comment and Comment 2.1).

Comment-4.3 *It is unclear why the STML behaviors are quite different at positive and negative biases for both molecules. What are the potential roles of inelastic electron tunneling or carrier injection mechanisms for different molecules at different bias polarities? Can the authors explain why the STML intensity is almost zero for Tol-Tpd-nNDI when the bias voltage exceeds -2 V (Fig. 3)?*

Reply: The experiment involves asymmetric junctions, as most STML experiments. It is a rather common feature, that light emission is not symmetric with bias voltage, even for simple plasmon emission without molecules. In our case, not only the contacts but even the molecule is not symmetric, not to speak of the differential conductivity. There is no reason to conclude that the behaviour needs to be symmetric. We are not aware of a complete explanation of this common feature in the literature (see e.g. Reecht et al. Phys. Rev. Lett. 112, 047403 (2014), Zhang et al. Nature 531, 623–627 (2016), Doppagne et al.

Science 361, 251-255 (2018), Doppagne et al. Nat. Nanotechnol. 15, 207–211 (2020), Rai et al. Phys. Rev. Lett. 130, 036201 (2023) and others) and thus prefer not to speculate here too much. We stick to our experimental observations.

Comment-4.4 *The dI/dV data is important for identifying the underlying excitation mechanism. Are the dI/dV data reproducible for different tip positions within a single molecule and for different molecular configurations? Are the STML intensity and dI/dV data shown in Fig. 3 collected on the same molecule and for the same tip position?*

Reply: In the data shown in Figure 3 for Tol-Tpd-nNDI, the STML intensity measurement and dI/dV spectroscopy (Figure 3a and 3b) were performed with the same tip on the same molecule. This allows their direct comparison. For the STS measurements, we show below spectra recorded at slightly different positions (gray spectrum is shown in the main text) for both molecules which show almost no position dependence (see Fig.R1 (left)). In the case of Tol-Tpd-sNDI, the data shown in Figures 3a and 3b were recorded on different molecules. However, dI/dV spectra do not depend on the position (see Fig.R1 (right)) (dI/dV measurements on different configurations were not performed).

Figure R1: Two spectra recorded at two slightly different pixels of **Tol-Tpd-sNDI** (left) and **Tol-Tpd-nNDI** (right).

Comment-4.5 *In the revised manuscript, the authors provide additional calculations to support the hypothesis of mechanical decoupling as the main reason for the observation*

of hot luminescence. However, it is not clear what kind of methods and basis sets are used for the calculation of the vibrational modes. For polyatomic molecules, there should be many vibrational modes and it is very common that the numerically calculated vibrational frequencies are not identical with the experimental observation. Thus, there could be several vibrational modes within a specific frequency range (e.g., around 485 cm^{-1}). To clearly identify the correspondence between the vibrational modes and the experimentally observed vibronic peaks, the vibrational-resolved emission spectrum should be calculated and compared with the experimental spectrum [see e.g., Fig. 2a in *Nat. Commun.* 12, 1280 (2021)]. The authors should try such calculations or at least calculate the Franck-Condon factors for the vibrational modes. Without these calculations, it is not convincing how the authors identify the five most intense vibronic peaks as the modes shown in Supplementary Movies 1–5.

Reply: The reviewer is indeed correct that there is some ambiguity in the identification of the five experimentally observed vibronic peaks. The two, which are discussed in more detail in the main manuscript, have been chosen according to the respective intensity of the calculated vibrational modes in the corresponding energy range, as listed in the following tables. Please note that DFT has a tendency to overestimate vibrational frequencies. For the vibron at 485 cm^{-1} , we therefore chose to investigate the range from $470\text{-}510\text{ cm}^{-1}$, where the following vibrations are found from density functional theory:

Tab. S1: List of vibrational modes of **Tol-Tpd-nNDI-Ac** ranging from $470\text{-}510\text{ cm}^{-1}$. All modes centered around $499\text{-}501\text{ cm}^{-1}$ are variations of the mode we depicted in the Supplementary Movie 2. The next intense vibration at 494 cm^{-1} is also clearly located at the tripodal scaffold.

Mode Nr.	Freq. [cm^{-1}]	Intensity [km/mol]
163	471.4	5.676
164	476.7	1.517

165	477.4	1.591
166	482.2	6.279
167	489.2	0.798
168	489.9	2.771
169	494.2	5.217
170	499.2	16.540
171	500.2	1.339
172	501.4	2.665
173	510.7	0.812

As can be seen, the most intense frequencies are centered around 499-501 cm^{-1} , which are all variations of the mode we depicted in the Supporting Movie 2. Another intense vibration at 494 cm^{-1} is also clearly located at the tripodal scaffold. The only vibration listed in Tab. S1 that is located at the NDI core is the one found at 482 cm^{-1} . This one is less intense than the cluster at 499-501 cm^{-1} and, in addition, DFT has a tendency to overestimate vibrational frequencies, making it very unlikely that this is the mode observed in the experimental spectra, which is located at 485 cm^{-1} .

Tab. S2: List of vibrational modes of **Tol-Tpd-nNDI-Ac** ranging from 1600-1730 cm^{-1} . These frequencies are all variations of the C=O stretching mode.

Mode Nr.	Freq. [cm^{-1}]	Intensity [km/mol]
497	1603.8	31.623
498	1607.5	17.187
499	1608.6	16.926
500	1613.5	7.165
501	1614.0	8.085
502	1616.6	11.677

503	1618.1	0.963
504	1618.8	31.833
505	1655.1	298.056
506	1662.8	131.490
507	1689.2	210.700
508	1696.1	114.087
509	1726.2	156.842
510	1727.8	167.284
511	1730.8	162.164

The batch of frequencies located in the range from 1600-1730 cm^{-1} are all variations of C=O stretching modes, where we again depict the most intense one at 1655 cm^{-1} in Fig. 5c. These two Tables S1 and S2 are now included in the SI. Furthermore, we improved the description of the theoretical methods used in the supporting information. It should now be clear what method has been used to obtain the vibrational spectrum.

Unfortunately, calculations as performed in Nat. Commun. 12, 1280 (2021) for pentacene (36 atoms, D^{2h} symmetry) are currently not possible for such extended systems as Tol-Tpd-sNDI (199 atoms, C^1 symmetry). The computational effort is on a completely different scale for our simulations. While an excited state optimization of pentacene with the methodology used in Nat. Commun. 12, 1280 (2021) is basically taking only a few minutes on a modern laptop, the excited state optimization of Tol-Tpd-sNDI at the highly advanced local hybrid density functional level of theory taking weeks on fully fledged compute clusters. Accessing the vibrational-resolved emission spectrum in the plasmonic junction is then little more than just state optimization, as also the $3n(-6)$ vibrational modes need to be assessed and furthermore processed. The vibrationally-resolved emission spectrum is therefore unfortunately not available at the moment because further technological advances, on both algorithmic and technical sides, will be required to perform such

calculations. Concerning the presented modes, it does not help to present the full number of vibrons in this paper and we restrict ourselves to only those that are intense.

Our experimental observations speak a clear language. They show that the HL peaks have indeed long lifetimes and their intensity is consistent with the measured Franck-Condon parameters. The calculations only serve to identify the modes and to rationalize the long lifetimes. A calculation of a vibrational resolved spectrum of such a large molecule in a plasmonic junction is a major undertaking and beyond our capabilities (and most likely also for our community). As stated before, this might change in the future with more powerful methods/computers. To allow reproduction of the vibrational modes we considered and to facilitate future full modeling, we included the full configuration of the relaxed molecule.

Comment-4.6 *In the revised manuscript, the authors “excludes the possibility of a cavity enhanced photon emission rate as the explanation for the HL peaks”. In my opinion, this conclusion should be treated with care. It is known that the lifetime of electronic excited states for a free-space molecule in vacuum is usually on the order of nanoseconds while the lifetime of vibrational excited states is on the order of picoseconds. Thus, even without the mechanical interaction between the molecule and the substrate, the vibrational excitation will decay much faster than the electronic excitation, making it very difficult to observe hot luminescence. The mechanical decoupling of the chromophore may increase the lifetimes of the chromophores’ vibrational modes, but this does not guarantee the occurrence of hot luminescence without the involvement of other factors (e.g., the Purcell effect that leads to the increase of the decay rate).*

Reply: The reviewer is right that the Purcell effect enhances radiative processes. The enhancement is due to a localized electric field in the plasmonic junction. This effect is active for all vibrations the same way, as the plasmon resonance is very broad and couples to the electronic part of the transition. This mode does not show sharp resonances at any of the energies in the range of interest. However, we observe that not all vibrational loss peaks are also found in HL. This simply excludes that the HL can be purely due to a Purcell effect. Of course, the letter is still active. We thus changed the sentence discussing

this to "This excludes the possibility of a cavity enhanced photon emission rate as the only explanation for the HL peaks."

Reviewer 5 (Remarks to the Author):

The authors have submitted a revised manuscript, in which they provided additional experimental data and theoretical calculations significantly improving the quality and generality of the paper. In particular, the extended discussion on the electronic structure, vibronic spectroscopy, and mechanical decoupling is very valuable. The main concerns raised by the referees have been answered. Some minor issues remain and I list them below. Once they are addressed, I will be happy to recommend the manuscript for publication in Nature Communications.

Comment-5.1 *My main concern in this version of the manuscript relates to the excitation mechanism. As of now, the authors consider only inelastic energy transfer. In the case of nNDI, this interpretation is reasonable for bias voltages between the onset of the emission and the onset of the NIR (2.36 V). Some data, however, are recorded at bias voltages higher than this onset, for example, the current dependence in Fig. S5 or the maps in Fig. 4, where one can expect that the excitation is realized by sequential charging and discharging of the molecule. While the inelastic tunneling excitation path still remains possible in principle, direct charging often is the dominating mechanism (see Phys. Rev. Lett. 122, 177401, 2019), which also is a one-electron process and the light intensity increases linearly with the current. In that respect, it is unfortunate that the bias dependence in the new Fig. 3 does not extend to voltages above the onset of NIR. That may also be the mechanism at play for the excitation of sNDI. Overall, this point needs more discussion.*

Reply: We thank the referee for this important point. Indeed, "Phys. Rev. Lett. 122, 177401, 2019" states that as soon as the NIR is reached, the peak intensity shows distinct

step-wise increases indicating that direct charging is the dominating mechanism. We simply do not observe this second step in the intensity excluding the mentioned mechanism by the criteria of that work. We were able to measure light intensities of nNDI up to a voltage well above the NIR (see in Fig.R2), and we have added this figure to the SI as requested by the reviewer. See also our reply to Comment 4.1 for the revised discussion of the excitation mechanism.

Figure R2: Integrated photon counts of Q_x emission line as a function of applied sample bias voltages. Parameters for recording the spectra are $I = 0.50$ pA, $t = 10$ s

Comment-5.2 The authors now show different spectra recorded on sNDI. As they argue in the reply to one of the comments (1.7), the interpretation of these measurements is complex due to the plasmonic response. Have the authors normalized the spectra by the respective plasmonic responses of the used tips? Such an operation could help understand the role of the enhancement in these measurements.

Reply: We have not normalized the spectra by the plasmonic response. In the energy range of the observed light in nNDI, the plasmonic response does not show any strong variation (see also SI, where we report the plasmonic response). For sNDI, the lines are close to the absorption edge of Au(111) and the plasmonic response indeed varies with energy. Nevertheless, the results of sNDI only serve to show that the light emission

can be chemically tuned. All the main claims of the work are based on the nNDI.

Comment-5.3 Line 226: “ Q_n and Q_s are absent for negative U ”. However, there is a signal visible for $V < -2$ V in the red curve in Fig. 3a. Is it of plasmonic origin? The authors should comment.

Reply: Yes, this relatively weak light emission is of plasmonic origin. We mention that now in the text: “the small photon counts in the negative bias are of plasmonic origin”

Comment-5.4 Lines 277-279: This sentence is not clear language-wise and should be revised.

Reply: We apologize for this grammatical mistake and have corrected the sentence, which now reads: “The STML spectrum recorded at $U = 1.98$ eV, i.e. at a bias voltage just above the photon energy of the main line Q_n , shows a fully developed spectrum together with the lower energy peaks (LE-band) (see blue spectrum in Fig. 5a).”

Comment-5.5. Lines 291-293: I am not sure I understand this sentence, that is, isn’t a vibron annihilated during emission, say from $S1v1$ to $S0v0$?

Reply: Yes, the referee is right, a vibron is annihilated during the emission process, but not during the excitation process as in the case of Anti-Stokes emission. We now clarify this and the revised sentence now reads: “Note that, since the HE-band peaks are not excited together with the main Q_n line, but only when the tunnelling electrons have the energy of the corresponding transition (see Fig. S4), the excitation process is not involving annihilation of vibrons (which cause anti-Stokes lines).”

Comment-5.6 Lines 295-299. I found this sentence contradicting, as one would consider experiments with PTCDA on NaCl typical experiments of a single molecule adsorbed on thin insulating layers. Also, reference 20 is a paper considering a self-decoupled system.

Reply: We now specify that the experiments with PTCDA are an exception to the typical experiments (see also Comment 2.2).

Comment-5.7 Related to the same paragraph, it would be helpful for the readers to explicitly clarify the difference between the vibrational hot luminescence and emission from

higher-lying excited states violating Kasha's rule. Besides the emission of H2Pc mentioned in comment 1.1, the emission from a much higher-lying state was reported in the case of porphyrins (Appl. Phys. Lett. 117, 243301, 2020).

Reply: The absence of vibrational deexcitation indeed breaks Kasha's rule and we do not claim otherwise. There are other examples of violations of Kasha's rule, but they are not the scope of this work. To discuss all the possible ways of breaking this rule and give examples and citations would take us over the length limit and would potentially confuse the reader.

Comment-5.8 *The supplementary movies should be labeled with the illustrated modes.*

Reply: We now label the Supplementary Movies with the illustrated modes. All files are named as follows: "Supplementary Movie X: Calculated animation of vibrational mode at YYYY cm⁻¹ of Tol-Tpd-nNDI"

REVIEWERS' COMMENTS

Reviewer #4 (Remarks to the Author):

The authors have made proper responses to the comments raised by the referees and have also revised the manuscript accordingly. Therefore, I now recommend its publication in Nature Communications.

Reviewer #5 (Remarks to the Author):

The authors have submitted a revised manuscript, which I am happy to recommend for publication in Nature Communications. As of now, two minor issues remain that should be addressed before publication.

1. Line 240: "For Tol-Tpd-nNDI, the NIR appears clearly above the energy of the main emission Q_n , excluding excitation via sequential charging and discharging of the molecule." - This sentence is unfortunately incorrect, as the excitation via sequential charging and discharging does require the NIR (or another state) to be above the emission line energy, see Phys. Rev. Lett. 130, 126202, 2023 and Phys. Rev. Research 5, 033027, 2023. I suggest removing that sentence.
2. Line 234/Fig. S5: A linear current dependence is expected for both sequential charging and inelastic energy transfer mechanisms. Therefore, while it supports the inelastic mechanism, it does not rule out sequential charging, especially since it is recorded above the NIR onset, and the authors should consider some rewording.

Overall, the lack of any feature around NIR in the light vs. bias dependence (Fig. S4) is quite interesting but at the same time, the tip likely retracts substantially (constant current measurement) when reaching the NIR energy, which may hinder some effects. It will be interesting to see further work on similar systems to develop a more profound understanding of light excitation in such self-decoupled structures.

Point-by-point response to the reviewers' comments:

We would like to thank all the reviewers for their detailed and careful evaluation of our revised manuscript entitled as "*Electrically and mechanically decoupled single chromophores by tripodal scaffolds*" and for supporting the publication. We hereby resubmit our final version, where we address all the remaining suggestions of Reviewer 5 point-by-point in our response below. In the text below, the reviewers' comments are set in italic font, while our answers, in blue, begin with the keyword "Reply:". The changes made in the revised manuscript are then written in red under the individual answers.

Reviewer #5 (Remarks to the Author):

The authors have submitted a revised manuscript, which I am happy to recommend for publication in Nature Communications. As of now, two minor issues remain that should be addressed before publication.

Comment 1: Line 240: *"For Tol-Tpd-nNDI, the NIR appears clearly above the energy of the main emission Q_n , excluding excitation via sequential charging and discharging of the molecule."* - This sentence is unfortunately incorrect, as the excitation via sequential charging and discharging does require the NIR (or another state) to be above the emission line energy, see *Phys. Rev. Lett.* 130, 126202, 2023 and *Phys. Rev. Research* 5, 033027, 2023. I suggest removing that sentence.

Reply: The referee is right that this sentence alone is not a valid statement in general. However, together with the information provided a few lines above, the conclusion is correct "For both **Tol-Tpd-nNDI** and **Tol-Tpd-sNDI** emission of Q_n and Q_s is observed as soon as the energy of the tunnelling electrons equals the energy of the emission lines Q_n and Q_s .", "An excitation via sequential charging and discharging of the molecule would result in a step-wise increase of the photon emission rate at a higher voltage¹², which we do not observe for voltages up to 3 V (for **Tol-Tpd-nNDI** see Suppl. Fig. 4)."

In order avoid any misunderstanding, we rephrase the sentence and now write:

For **Tol-Tpd-nNDI**, the NIR with an onset of $U = 2.36$ V appears clearly above the threshold voltage of 1.95 V for the onset of the main emission line Q_n (with an energy of 1.95 eV), excluding excitation via sequential charging and discharging of the molecule at voltages up to 2.36 V. The onset of an excitation via sequential charging and discharging of the molecule would result in a step-wise increase of the photon emission rate around this voltage¹², which we do not observe for voltages up to 3 V (for **Tol-Tpd-nNDI** see Suppl. Fig. 4).

Comment 2: Line 234/Fig. S5: *A linear current dependence is expected for both sequential charging and inelastic energy transfer mechanisms. Therefore, while it supports the inelastic mechanism, it does not rule out sequential charging, especially since it is recorded above the NIR onset, and the authors should consider some rewording.*

Reply: As the referee correctly states, a linear current dependence is expected for and supports an inelastic energy transfer mechanism. This is exactly what we write in the main text: “This (*inelastic energy transfer*) is in full agreement with STML experiments at different tunnelling currents”, and in the SI: “This supports the argument of the light emission being driven by inelastic energy transfer via plasmons....”

In our manuscript, we clearly do not claim that the observation of a linear current dependence rules out sequential charging and therefore see no reason for rewording.

Comment 3: *Overall, the lack of any feature around NIR in the light vs. bias dependence (Fig. S4) is quite interesting but at the same time, the tip likely retracts substantially (constant current measurement) when reaching the NIR energy, which may hinder some effects. It will be interesting to see further work on similar systems to develop a more profound understanding of light excitation in such self-decoupled structures.*

Reply: The referee is technically correct that there's always a possibility that another excitation mechanism is involved but not apparent in the data because it's obscured

by other effects. However, we prefer not to speculate and stick to the simplest hypothesis that explains our observations.